# Probing the Brain–Body Connection Using Transcranial Magnetic Stimulation (TMS): Validating a Promising Tool to Provide Biomarkers of Neuroplasticity and Central Nervous System Function

**DOI:** 10.3390/brainsci11030384

**Published:** 2021-03-17

**Authors:** Arthur R. Chaves, Nicholas J. Snow, Lynsey R. Alcock, Michelle Ploughman

**Affiliations:** L.A. Miller Centre, Recovery and Performance Laboratory, Faculty of Medicine, Memorial University of Newfoundland, St. John’s, NL A1A 1E5, Canada; ar.chaves@hotmail.com (A.R.C.); njsnow@mun.ca (N.J.S.); lynsey.alcock@gmail.com (L.R.A.)

**Keywords:** multiple sclerosis, transcranial magnetic stimulation, biomarker, corticospinal excitability, walking speed, hand function, walking function, cognition, fatigue, neuroplasticity

## Abstract

Transcranial magnetic stimulation (TMS) is a non-invasive method used to investigate neurophysiological integrity of the human neuromotor system. We describe in detail, the methodology of a single pulse TMS protocol that was performed in a large cohort of people (*n* = 110) with multiple sclerosis (MS). The aim was to establish and validate a core-set of TMS variables that predicted typical MS clinical outcomes: walking speed, hand dexterity, fatigue, and cognitive processing speed. We provide a brief and simple methodological pipeline to examine excitatory and inhibitory corticospinal mechanisms in MS that map to clinical status. Delayed and longer ipsilateral silent period (a measure of transcallosal inhibition; the influence of one brain hemisphere’s activity over the other), longer cortical silent period (suggestive of greater corticospinal inhibition via GABA) and higher resting motor threshold (lower corticospinal excitability) most strongly related to clinical outcomes, especially when measured in the hemisphere corresponding to the weaker hand. Greater interhemispheric asymmetry (imbalance between hemispheres) correlated with poorer performance in the greatest number of clinical outcomes. We also show, not surprisingly, that TMS variables related more strongly to motor outcomes than non-motor outcomes. As it was validated in a large sample of patients with varying severities of central nervous system dysfunction, the protocol described herein can be used by investigators and clinicians alike to investigate the role of TMS as a biomarker in MS and other central nervous system disorders.

## 1. Introduction

Biological markers (‘biomarkers’) are surrogate markers of disease activity used in the diagnosis, characterization, prognostication, and surveillance of disease throughout its natural history and in response to therapy [1]. Certain biomarkers provide information on neurophysiological processes related to neurologic disease, which allows researchers to predict potential for disease recovery and understand mechanisms of prospective treatments. This is critical information needed to determine how neurorehabilitation may alter neuroplasticity in heterogeneous disorders of the central nervous system such as stroke and multiple sclerosis (MS) [2,3,4,5]. Some biomarkers can be collected and assayed from biological fluid such blood [6,7] or cerebrospinal fluid [7], while others could involve sophisticated imaging techniques such as positron emission tomography or magnetic resonance imaging (MRI) [3,8,9]. One potential method to gather biomarkers of central nervous system (dys)function and neuroplasticity is via Transcranial Magnetic Stimulation (TMS) [10].

### 1.1. Overview of TMS Methods

In the human central nervous system, the degree of neuronal *excitation* is mediated by glutamatergic neurotransmission while neuronal *inhibition* is mediated by gamma-aminobutyric acid (GABA)ergic systems [10,11]. Neuronal excitation and inhibition can be measured by investigation of corticospinal excitability (CSE) using TMS [10] (Figure 1). Many groups internationally use TMS to probe central nervous system function; however, methods often vary considerably among laboratories, making it difficult to compare findings across studies. It is also not clear which TMS techniques (e.g., single pulse or paired pulse) and which TMS variables are relevant as rehabilitation and neuroplasticity biomarkers.

#### 1.1.1. Motor Thresholds

The most common TMS variable used to investigate CSE is the motor threshold. Motor thresholds represent the lowest TMS intensity required to elicit a motor evoked potential (MEP), either during complete muscle relaxation (resting motor threshold; RMT) or slight tonic muscle contraction (active motor threshold; AMT) [10]. Motor thresholds reflect the strength and size of the most excitable elements of the muscle representation in the primary motor area, which include the availability of excitatory neurotransmitters (e.g., glutamate) and their receptors and function of ion channels in cortical and spinal neuron populations [10]. Inability to obtain a MEP during TMS suggests poorer integrity of the corticospinal tract [12]. In this way, an increase in motor threshold (i.e., the primary motor area requires higher stimulation intensity to elicit a MEP) indicates decreased CSE.

#### 1.1.2. Excitatory Recruitment Curve

A more thorough TMS protocol to investigate neuronal availability and strength of excitatory neurotransmission involves examining recruitment curves. The recruitment curve requires incrementally increasing TMS stimulus intensities to examine corresponding increases in MEP amplitudes that result from faster temporo-spatial summation at cortico-motoneuronal synapses [10]. The excitatory recruitment curve (eREC), also referred to as the MEP input-output curve or stimulus-response curve [13], indexes the excitability of the least to most excitable neuronal populations in a motor representation, using TMS stimulus intensity × MEP size plots [14] (Figure 2). Various input-output properties of the human neuromotor system can be derived from the eREC, including the recruitment gain and accuracy of corticospinal neurons (from the slope and *R-squared* (R^2^)) [14,15,16], and total excitability [17,18] of the corticospinal pathway (area under the recruitment curve (AUC) calculations). Previous work has demonstrated reduced CSE gain (eREC slope) and overall excitability (eREC AUC) in stroke survivors, as well as associations between these eREC parameters with central nervous system damage beyond the primary motor area [19]. As for accuracy (eREC R^2^), previous authors have proposed that values below 0.7 may reflect insufficient ability of the brain to appropriately recruit neurons [17,19].

Evidence suggests there is consistency across the various outcomes derived from the eREC [20], so they can be reliably measured with few stimulations [21]. Such efficiency supports a more minimalist approach to TMS data collection which may be appropriate in clinical populations who can be prone to testing fatigue.

#### 1.1.3. Cortical Silent Period (CSP)

The CSP is the brief period of quiescence in target muscle electromyographic (EMG) activity following TMS-evoked cortical stimulation (Figure 1C). Whereas MEP amplitude and eREC properties are markers of corticospinal excitability, the duration of the CSP indicates GABAergic-mediated corticospinal inhibition. Corticospinal inhibition is an important parameter to examine because excessive inhibition is associated with a blunted capacity for long-term potentiation (LTP), a fundamental building block of neuroplasticity underlying the formation of new central nervous system pathways in learning and rehabilitation [22,23]. Short- and long-lasting CSPs are believed to be mediated by GABA_A_- and GABA_B_-receptor activity, respectively [10]. The exact structural and functional mechanisms—including cortical versus spinal contributions—that underlie CSP propagation represent an area of intense scrutiny across the literature [24] and is discussed in other work [25,26].

#### 1.1.4. Inhibitory Recruitment Curve

Similar to the eREC, which describes the properties of excitatory (glutamatergic) corticospinal neuron populations, an inhibitory CSP recruitment curve (iREC) can be utilized to index the activity of inhibitory (GABAergic) interneurons [26] (Figure 2). Additionally, while other TMS protocols such as short-(SICI) and long-interval intracortical inhibition (LICI) and short-interval intracortical facilitation are also thought to be proxies of GABAergic activity in the central nervous system, they are beyond the scope of the current work and are reviewed elsewhere [10,27].

#### 1.1.5. Transcallosal Inhibition and the Ipsilateral Silent Period

Another clinically useful TMS-derived variable describes transcallosal inhibition. By applying a suprathreshold (i.e., above motor threshold) TMS stimulus over the target motor representation while the ipsilateral target muscle is contracting, the duration of interruption of the EMG activity in the active muscle captures the ipsilateral silent period (iSP) [27]. The iSP is thought to be a measure of interhemispheric or transcallosal inhibition [28], the influence of one cerebral hemisphere’s activity over the other via projections across the corpus callosum [25]. The iSP also provides a proxy measure of the activity of cortical [28] glutamatergic and GABAergic neurons [29]. Interhemispheric or transcallosal inhibition, quantified by the size and duration of the iSP [25], may modulate brain changes and recovery after stroke [30] and is related to indices of disease severity and progression in MS [31,32].

Given the impact that neurological damage and neurodegeneration has on glutamatergic (excitatory) and GABAergic (inhibitory) central nervous system mechanisms [30,33], as well as performance of functional motor tasks involving coordination of activity between cerebral hemispheres [34], it is prudent to investigate CSE and inhibitory corticospinal mechanisms using TMS. This tool could help better understand how, and to what extent, the lesion-disrupted central nervous system can undergo recovery through neuroplasticity [30,33] and help the development of better CNS-targeted treatments (e.g., non-invasive brain stimulation [35]) for MS.

However, more research is required to examine its validity in large samples of people with neurological disorders [2], including MS [27]. 

Multiple sclerosis is a progressive autoimmune disease typically diagnosed in children and young- and middle-aged adults that is characterized by both chronic neurodegeneration and sudden intensifications of neuroinflammation that induce demyelination and neuronal death [36,37,38,39]. Spontaneous recovery from relapses (i.e., remitting phase) is often partial, leading to disability progression over time [36]. Indeed, approximately 80% of people with relapsing-remitting MS (RRMS) will develop secondary progressive MS (SPMS) over their disease course, a phase in which there is a steady progression with few distinct relapses [37,38]. Further, 10–15% of people diagnosed with MS experience primary progressive MS (PPMS), a steady progression of symptoms with no remissions or relapses [38,40]. Although these three forms of MS (i.e., RRMS, SPMS, and PPMS) have been part of the lexicon of MS for decades, new understanding of disease activity, lesion formation, and gray matter atrophy has led to reconsideration of these labels [39,41]. Recent recommendations suggest that MS could be better categorized as active or inactive based on relapses and disease activity seen on central nervous system imaging [42]. This new perspective on MS clinical phenotypes [39] indicates that the MS spectrum is more complex than previously thought. In all cases, people with MS develop a variety of autonomic (e.g., thermoregulatory, sexual, and urinary dysfunction) [43], physical (e.g., fatigue, weakness), and cognitive (e.g., memory and learning impairments) deficits [44,45,46,47] which negatively influence all dimensions of quality of life [44,46,47,48,49]. 

Thus, variables derived using TMS could help to: (1) provide a better understanding of neuropathophysiological events involved in the onset and sequelae of MS [32], (2) predict disease severity and progression for prognostication purposes, and (3) investigate whether rehabilitation therapies [50] and pharmacological treatments [5] are truly acting on the central nervous system to enhance neuroplasticity and recovery [27]. Although recent work has recognized the potential role of TMS in providing biomarkers of central nervous system functioning in MS, conclusions were based on small studies that demonstrated a high risk of bias [27]. Additionally, many TMS procedures require specialized equipment, technical expertise and, in most cases, long testing times. With numerous existing TMS protocols and paradigms, it is important to select and validate a core-set of TMS variables that are sensitive to changes in clinical outcomes and are time-efficient to collect. 

Therefore, the main objective of this study was to test and describe in detail the methodology of a single pulse TMS protocol that was performed in a large cohort of people with MS. There were three research questions:As administering TMS in both brain hemispheres doubles the testing time, is it necessary to collect TMS variables bilaterally? To what extent do TMS variables correlate with clinical outcomes, specifically, motor (i.e., walking speed and nine-hole peg test (9HPT)) and non-motor function (i.e., fatigue and cognitive status)?Of the more than 25 variables derived from this TMS protocol, which variables are most strongly associated with severity of MS symptoms (controlling for confounding variables) and should be considered as part of a core-set?

## 2. Materials and Methods

### 2.1. Participants

Study ethics approval was granted by the Human Research Ethics Board (HREB #20161208) at Memorial University of Newfoundland. Informed consent was gathered from participants in accordance with the Declaration of Helsinki. Participants with a confirmed diagnosis of MS, according to the revised McDonald Criteria [51], were sequentially recruited through the provincial MS Clinic during their scheduled neurologist visit. Participants’ Expanded Disability Status Scale (EDSS) [52] value was extracted from the medical chart record of the visit. We attempted to recruit at least 100 patients that were representative of a typical MS clinical sample. Prospective participants were included if they provided written informed consent and agreed to attend a separate 2-h visit in the same hospital for walking, hand dexterity, fatigue, cognitive, and TMS testing. Participants completed a standard TMS safety screening form [53]. In some cases, affirmative answers, such as taking anti-depressant drugs, were not considered to represent risk and/or absolute contraindication to single pulse TMS assessment [53]. The absolute exclusion criteria included the presence of any metallic hardware close to the TMS coil, for instance medical pumps or cochlear implants [53].

### 2.2. Clinical Testing Procedures

The visit began with measurement of height and weight and the collection of demographic information. Participants then completed the symbol digit modalities test (SDMT): a validated tool used to measure information processing speed in people with MS [54,55]. Briefly, the participant was given 90 s to match, in writing, specified numbers to a set of geometric figures presented in an answer key. Fewer correct answers are indicative of slower processing speed. Next, participants were asked to mark a line along a 100 mm visual analogue scale to indicate how much their MS-related fatigue affected their daily life and relationships from ‘no effect at all’ (0 mm) to a ‘very big effect’ (100 mm). Participants then completed bilateral upper extremity dexterity testing using the 9HPT [56]. The 9HPT is a valid and reliable method to measure gradients of upper limb impairment in MS [57]. Participants transferred wooden pegs one at a time from a container to holes in a wooden block and then removed them again, with the time taken to complete the activity recorded in seconds. Participants completed the task using the dominant hand first, followed by the non-dominant hand; this was repeated and the score for each hand was averaged [56]. Participants then completed pinch strength (B & L Engineering, Santa Ana CA, USA) and grip strength (Jamar Dynamometer, Lafayette Instrument Corporation, Lafayette, IN, USA) testing using calibrated dynamometers, while seated without back or arm support, alternating sides, twice on each hand and averaged to obtain maximum values. Pinch and grip strength were used to determine participants’ weaker and stronger hands for the purposes of TMS testing [58]. Thereafter, TMS testing took place. This was followed by measurement of self-selected walking speed using a 4-metre-long instrumented walkway (Protokinetics, Havertown PA, USA). Footfalls were extracted from the walkway to derive speed in cm/s. TMS testing preceded walking to avoid possible acute effects of walking on CSE [59]. Walking speed (cm/s) was corrected for participants’ height in cm (cm/s/height_cm_) to account for the relationship between height and walking speed [60].

### 2.3. TMS Testing Procedures

#### 2.3.1. Electrode Placement and Skin Preparation

Although the target muscle varies between studies, we chose the first dorsal interosseous (FDI) muscle to measure TMS-evoked MEPs because of the muscle’s large motor cortex representation and smaller motor unit-to-muscle fiber innervation ratio [61,62]. This is especially important and practical when testing participants with greater levels of neurological disability, who may have high motor thresholds [63,64,65] and low, difficult-to-measure, MEP amplitudes [50,58,59]. This allows for less participant discomfort and measurement of a greater number of TMS intensities on the recruitment curve [13]. The FDI muscle is also used in various functional fine motor activities of the hand, which makes studying the excitability of its motor cortical representation clinically relevant [66]. Previous studies confirm that TMS variables derived from upper extremity MEPs are representative of overall CSE [67,68]. Furthermore, the FDI is a commonly studied muscle representation in TMS studies of clinical populations (e.g., stroke [2,30], MS [2,27], amyotrophic lateral sclerosis, Parkinson’s disease, and Huntington’s disease [69,70]). The skin on both hands was prepared at the outset to minimize set-up delays during TMS testing. Six areas (three on each hand), slightly larger than the 4 cm EMG recording electrode (Conductive adhesive hydrogel foam electrode 3.5 cm × 4 cm, Coviden, Mansfield, MA, USA), were shaved, abraded with sandpaper, and wiped with 70% isopropyl alcohol swabs to reduce electrical impedance and improve signal to noise ratio. These six areas included the skin overlying the belly of the FDI muscle (active electrode), the ulnar styloid process (ground electrode), and the proximal interphalangeal joint of the index finger (reference electrode: Figure 1D). When the participant’s ulnar styloid process was difficult to palpate, the ground electrode was relocated to the medial epicondyle or olecranon process of the elbow, or another prominent bony landmark on the same side.

#### 2.3.2. TMS System

A Magstim BiStim 200^2^ magnetic stimulator (Magstim Co., Whitland, UK) connected to a double 70 mm figure-of-eight coil (Magstim, Co.) was used to deliver monophasic magnetic pulses over the hand area of the primary motor cortex of each hemisphere separately. During all procedures, the TMS coil was held tangentially to the scalp with the handle pointing backward and laterally at an angle of 45° from the midline and perpendicular to the central sulcus (Figure 1A). This position delivers posterior-to-anterior directed TMS pulses that produce large MEPs resulting from the gradual recruitment of I-waves [71,72]. The hemisphere tested first (corresponding to the weaker or stronger upper extremity) was randomly assigned. We obtained the motor hotspot, motor thresholds, recruitment curves, and transcallosal inhibition values from one hemisphere/hand entirely before switching to the opposite side. Neuronavigation using Brainsight™ (Rogue Research, Montreal, QC, Canada) guided the TMS coil position. Brainsight™ was also used to sample and store MEPs and silent periods with its built-in EMG system. This system uses a 2500 V/V amplification, a 3 kHz sampling rate, a gain of 600 *V*/*V* and a bandwidth of 16–550 Hz. The Montreal Neurological Institute brain template was rendered into the Brainsight™ software, which we used as a 3-D stereotaxic template to guide coil position [73,74]. During each TMS trial, EMG data were collected in a 900 ms sweep (100 ms pre-TMS stimulus to 800 ms post-stimulus). To avoid overheating and subsequent automatic shutdown of the TMS stimulator or coil, we positioned the unit close to an air conditioner and, when necessary, rested the coil on a cooling pad between trials.

#### 2.3.3. Calibration

Guided by the Brainsight™ neuronavigation software, we first calibrated the position of the coil and the position of the participant’s head in 3-D space using an infrared camera and template image of the representative brain. The coil was outfitted with infrared markers and calibrated using a standard coil calibration plate. With the participant sitting in a custom-designed chair with headrest and armrests, infrared marker-equipped glasses were placed on the participant and a similarly equipped pointer was used to identify key anatomical markers (nasion, right and left tragus of the ears). The participant continued to wear the infrared marker-equipped glasses for the remainder of the TMS testing to map the position of the coil/stimulus.

#### 2.3.4. Obtaining the Motor Hotspot

We overlaid a 6 cm × 7 cm virtual grid over the area corresponding to the hand knob of each motor cortex to guide the creation of the individual’s motor map [75] (Figure 3). Individualized motor maps were used to aid in the localization of participants’ FDI motor cortical representation (motor hotspot) for each hemisphere. Motor maps are a reliable TMS method to determine the motor hotspot, because the techniquereduces inter-experimenter subjectivity and variability [33]. The first step was to position the TMS coil tangential to the participant’s scalp, roughly over the cortical representation for the contralateral hand (~2 cm lateral and 1 cm anterior to the vertex [13]). At this standard location, the aim was to first deliver a suprathreshold (i.e., above motor threshold) TMS stimulation (measured in % maximum stimulator output, % MSO) that elicited ~500 µV amplitude MEPs. Then, every grid target was stimulated (two to three times each) using the same % MSO and site (i.e., grid target). Stimulations were delivered at random inter-trial intervals ranging between 3–5 s to prevent any influence of systematically stimulating at the same frequency [76]. Using this technique, the site with the highest averaged MEP peak-to-peak amplitude was taken as the motor hotspot.

#### 2.3.5. Motor Thresholds

Motor thresholds were used to determine participants’ baseline level of CSE. RMT and AMT were defined as the minimum TMS intensity (% MSO) necessary to elicit at least five out of 10 MEPs with a peak-to-peak amplitude of ≥50 µV during muscle relaxation (RMT) and ≥200 µV during 10% of maximal voluntary contraction (AMT) [10], respectively. RMT was typically collected before AMT to prevent any influence of sustained or repeated tonic hand muscle contractions on CSE at rest. During RMT assessment, in cases where participants demonstrated inability to rest (evidenced by muscle EMG background activity), we would: (1) inform participants to relax as much as possible; (2) ask participants to perform a strong grip and then completely relax or, if needed; (3) collect AMT first, and resume RMT assessment later. Participants rested with their forearms on a pillow in their lap to avoid tension of shoulder and arm muscles during CSE assessment.

#### 2.3.6. Recruitment Curves

We obtained eREC and iREC data simultaneously during a 10% of maximal muscle contraction of the FDI (achieved using a pinch dynamometer). Voluntary muscle contraction reduces the TMS motor threshold and permits a greater range of suprathreshold stimuli to be delivered without exceeding the capacity of the TMS device (i.e., >100% MSO). Moreover, collection of data during voluntary muscle contraction provides information about the neuronal activity engaged during motor practice [77], reduces between-participant variability [58,78], and detects age- [77] and disease-related CSE changes [58,79,80] that are not apparent with resting measurements. An additional benefit of obtaining recruitment curve data while contracting the muscle of interest is the simultaneous collection of the CSP for each incremental increase in stimulator output [81] (Figure 2). With the participant contracting at 10% of maximal voluntary contraction (using pinch gauge), five to six stimulation trials, at random inter-trial intervals between 3–5 s apart, at 105%, 115%, 125%, 135%, 145%, and 155% of AMT (30–36 stimulations in total) were delivered in randomized order to prevent changing CSE while measuring it [77]. To ensure consistency, we entered the AMT value into Microsoft Excel (Redmond, WA, USA) to calculate the TMS intensities and to block-randomize the TMS intensities (105–155% of AMT).

#### 2.3.7. Transcallosal Inhibition

Obtaining transcallosal inhibition (quantified by iSP) was the last TMS experiment to be completed before moving on to the other hemisphere. The participant was asked to perform a maximum pinch grip using the hand ipsilateral to the hemisphere being stimulated (i.e., pinch gauge switched to the opposite side from recruitment curve assessment) and five to six ipsilateral TMS stimulations were delivered at 180% of RMT over the ipsilateral motor hotspot [82]. To illustrate, if the participant’s RMT in the right hemisphere was 50% MSO, they were asked to pinch the dynamometer at their maximum ability with their right hand while the TMS was fired over the right hemisphere using a stimulator intensity of 90% MSO. These parameters were chosen to correspond with maximal transcallosal and corticospinal neuron recruitment [10,83] with the intention of producing stable measurements of the iSP [84,85]. For measurement of iSP, trials were each separated by randomized inter-trial intervals of 10–15 s to mitigate fatigue in the participant [83]. If the required stimulus to examine iSP surpassed the limit of the stimulator (i.e., 1.8 × RMT > 100% MSO), the iSP experiment continued using 100% MSO. This was the case for three participants when measuring the hemisphere corresponding to the weaker hand, and four participants when measuring the hemisphere corresponding to the stronger hand.

Following completion of TMS collection in the first hemisphere (corresponding to weaker or stronger hand, in random order), all TMS measurements were repeated in the opposite hemisphere. After the entire experiment was completed, all data were transferred to Signal Software (v.6.0; Cambridge Electronic Design Limited, Cambridge, UK) and Microsoft Excel for post-processing and data reduction.

### 2.4. TMS Post-Processing and Data Reduction

#### 2.4.1. Motor Thresholds, MEP Latency, and AMT Symmetry

The AMT and RMT were recorded as % MSO (0–100%). Although AMT and RMT data did not require further processing, all MEPs were visually inspected to validate whether motor thresholds were collected as defined above. The EMG traces of each participant’s MEP trials (including 100 ms pre-stimulus to 800 ms post-stimulus) were inspected on a trial-by-trial basis. If there was a visible artifact compared to other MEPs, or excessive pre-stimulus background EMG activity (or presence of CSP in the case of RMT assessment), compared to other trials, the MEP trial was discarded [86,87]. Less than 1% of trials were omitted from the dataset on this basis.

Onset latency of MEPs was calculated as the time in milliseconds (ms) from the TMS stimulus to the MEP onset from all motor threshold trials in which there were MEPs (Figure 4). Resting and active MEP latencies were extracted from the MEPs recorded during RMT and AMT assessments, respectively. The MEP onset occurred when the MEP exceeded >±2 standard deviations (SD) from the EMG background activity. As MEP latency is influenced by height and limb length [88,89], these values were normalized by participant’s height in cm (ms/height_cm_).

In addition to describing TMS intensities used to elicit AMT and RMT, as well as active and resting MEP onset latencies, we also determined participants’ interhemispheric AMT and RMT asymmetries. The asymmetry ratio was calculated by dividing the motor thresholds of the hemisphere corresponding to the weaker hand by the motor thresholds of the stronger side (AMT asymmetry = AMT_W_/AMT_S_; RMT asymmetry = RMT_W_/RMT_S_) [58]. Values <1.0 indicate that the weaker side of the body has higher CSE, and values > 1.0 indicate that the weaker side has lower CSE. Our past work has shown that asymmetry of motor thresholds is related to various clinical outcomes in MS [58].

#### 2.4.2. MEP Amplitudes and eREC Parameters

Peak-to-peak MEP amplitudes (µV) were averaged across all trials for each TMS intensity (i.e., 105–155% of AMT). The linear relationship between the MEP amplitudes (µV) and TMS intensities (105–155% of AMT) was used to describe the eREC [32,90] (Figure 2). Here, the slope and R^2^ of the linear relationship, respectively, quantified the excitatory corticospinal tract recruitment gain and accuracy [10,19,32]. Overall excitability of the corticospinal tract was calculated as the AUC using the trapezoid integration rule ∆*X* × *(Y*1 + *Y*2)/2, whereby *X*-values were the TMS intensities used (i.e., X-axis values corresponding to 105–155% of AMT) and *Y*-values were the recorded MEP amplitudes (i.e., Y-axis values corresponding to MEP amplitude in µV) [17,18]. It is important to note that, when intensities below motor threshold (i.e., subthreshold; <100% of RMT or AMT) are performed and followed by suprathreshold intensities that are meant to force a neuronal recruitment plateau (e.g., >155% of motor threshold), the sigmoidal function (Boltzmann’s equation) would be preferred to calculate the gain (slope) and accuracy (R^2^) of the REC [19] (Figure 2). However, we used only TMS intensities between 105% and 155% of AMT, which correspond to the ascending portion of the eREC and therefore could be modeled with a linear regression equation [13]. This approach has been validated in other work [91,92]. 

#### 2.4.3. CSP Duration and iREC Parameters

The CSP onset began when the MEP amplitude surpassed >±2 SD from the mean pre-stimulus EMG background activity (i.e., 100 ms prior to the TMS stimulation) [10,26] (Figure 4). The end of the CSP was the point at which the EMG background activity post-MEP returned to within ± 2 SD from background EMG activity [10,26]. Using CSP onset and offset, we calculated CSP duration (i.e., *CSP duration* = *CSP offset* − *CSP onset* in ms) [10,26]. As well, similar to overall excitability above, overall inhibition was calculated as the AUC using the trapezoid integration rule ∆*X* × (*Y*1 + *Y*2)/2, whereby *X*-values were the TMS intensity used (i.e., X-axis values corresponding to 105–155% of AMT) and *Y*-values were the recorded CSP duration (i.e., Y-axis values corresponding to CSP duration in ms). We did not examine the slope or R^2^ of the iREC because these outcomes are not commonly described in the literature [10] and there is little evidence of their relevance in MS [50].

#### 2.4.4. Transcallosal Inhibition

Mean baseline EMG activity was used as the threshold to define the iSP [93]. Similar methods of iSP analysis have shown good inter- and intra-rater reliability [94]. For each participant, EMG data were full-wave rectified and averaged across all trials in each hemisphere separately. Mean pre-stimulus EMG amplitude, over the period of 100 ms prior to TMS delivery, was considered the baseline muscle activity (Figure 5) [93]. We considered several characteristics of the iSP including onset latency, duration, depth, and AUC. Onset latency was defined as the time in ms from the TMS stimulus to the iSP onset [95]. The iSP onset was the time point at which the average EMG activity clearly (by visual inspection comparing the average rectified EMG waveform to the mean pre-stimulus EMG from the same trials) fell below mean pre-stimulus EMG amplitude [95] for at least five consecutive data points (given EMG data were sampled at 3 kHz, this corresponds to ≥1.67 ms of data [96]). Similarly, iSP offset was the time point at which the average EMG activity clearly returned to mean pre-stimulus amplitude [95] for at least five consecutive data points (≥1.67 ms) [25,96]. The iSP duration was the time elapsed between the onset of the iSP to the offset. Depth was calculated as the average EMG amplitude during the iSP (µV), normalized as a percentage of mean pre-stimulus EMG amplitude (µV) [97]. The iSP AUC was determined via trapezoid integration as the area under the reduced EMG activity during the iSP using the equation ∆*X* × (*Y*1 + *Y*2)/2, whereby *X*-values were the iSP duration (i.e., X-axis values corresponding to time in ms during the iSP) and *Y*-values were the iSP depth (i.e., Y-axis values corresponding to depth of the iSP in µV) [98]. The iSP AUC was normalized as a percentage of pre-stimulus EMG area defined over a duration equal to the iSP using the trapezoid integration rule [98]. To consider the combined effect of iSP depth and duration we calculated the *iSP depth* × *duration* by multiplying normalized iSP depth (% of pre-stimulus EMG amplitude) by iSP duration (ms) [25].

### 2.5. Statistical Analysis

All statistical analyses were performed in SPSS (V26.0, IBM Corporation, Armonk, NY, USA). Statistical significance was set at *p* < 0.05. For all analyses, TMS variables included (1) motor thresholds: RMT, AMT, and RMT and AMT asymmetry ratios, (2) MEP latencies: resting and active MEP latencies (height corrected (ms/height_cm_)), (3) eREC: MEP amplitudes at 105–155% of AMT, AUC, eREC slope, and eREC R^2^, (4) iREC: CSP duration at 105–155% of AMT, iREC AUC, and (5) transcallosal inhibition: iSP onset latency, iSP duration, iSP depth, iSP AUC, and iSP *depth × duration.* For all analyses, clinical (motor and non-motor) outcomes of MS symptom severity included walking speed, 9HPT weaker hand, 9HPT stronger hand, fatigue, and SDMT.

#### 2.5.1. Differences in TMS Variables between Hemispheres

Across the MS literature, there is no consensus as to how TMS variables from each cerebral hemisphere are analyzed and presented. Some studies average data across hemispheres whereas others separate hemisphere data based on hand dominance or strength reported by participants [27]. We aimed to discern whether there were differences between the cerebral hemispheres corresponding to participants’ stronger vs. weaker hand, as this concept of asymmetry may be clinically relevant in MS [58]. To determine whether there were significant differences between hemispheres necessitating them to be evaluated separately, we first performed parametric paired-samples *t*-tests (test statistic reported as t_(degrees of freedom)_) or non-parametric Wilcoxon Signed-Ranks test (for non-normally distributed data, test statistic reported as z) [99] for all TMS variables outlined above. We also completed the same analysis with participants separated in groups based on a higher (EDSS ≥ 3) or lower (EDSS < 3) level of MS-related disability [100].

#### 2.5.2. Examining Relationships between TMS Variables and Clinical Outcomes

Pearson’s correlations (test statistic reported as r) were performed between TMS variables and clinical outcomes to determine which TMS variables were most strongly related to which clinical outcomes. The size of the correlation coefficient (r) was interpreted as ‘zero’ 0, ‘weak’ 0.10–0.39, ‘moderate’ 0.40–0.69, ‘strong’ 0.70–0.99, and ‘perfect’ 1.0 [101]. 

All non-normally distributed data were transformed to both perform Pearson’s correlations and meet the assumption of normality required for regression analysis (see below). Moderately and strongly skewed data were square-root transformed or logarithmic-transformed, respectively. Extremely positive skewed data were transformed by dividing by 1 [102]. Post-transformation, normality was assessed again to ensure that the distribution had, in fact, become normal.

#### 2.5.3. Determining Strongest Predictors of Clinical Outcomes

To determine whether there was a core set of TMS variables that best predicted MS symptom severity, we fit five separate hierarchical regression models (one for each of the five dependent variables, as listed above) using TMS variables as predictors and MS clinical outcomes as dependent variables. 

First, we included age and sex to determine how much of the variability in the dependent variables was accounted for by these potential confounding factors. Only TMS variables and dependent variables that were significantly correlated were entered into the model, except in cases where the TMS variables were highly correlated with one another (i.e., r ≥ 0.70). In these cases, we only included the TMS variables with the strongest correlation to the dependent variables. During the regression, the presence of multicollinearity was also checked with tolerance and variance inflator factor (VIF) values (<0.1 and >10.0, respectively) [103]. The order in which the TMS variables were added was based on the Pearson’s correlation results; TMS variables with strongest correlation with the dependent variable (MS symptoms severity) were added first. Independence of observations was tested with Durbin-Watson test (~2.0) [104] and strength of the relationship was evaluated using R^2^ [105]. The size of the coefficient of determination (regression R^2^) was interpreted as ‘very weak’ < 0.3, ‘weak’ 0.3–0.49, ‘moderate’ 0.5–0.69, and ‘strong’ > 0.7 [106]. To investigate whether a linear relationship between the dependent and the independent variables existed and whether the assumption of homogeneity of variance (i.e., homoscedasticity) was met, a scatterplot of the studentized residuals against the unstandardized predicted values from the regressions was plotted. Possible outliers and influential points were identified with Cook’s distance > 1.0 [107] and leverage values > 5.0 and were excluded from the regressions [108,109]. The assumption of normality of the residuals during the regressions was checked with normal Q-Q plots of the studentized residuals [105,110].

#### 2.5.4. Use of CNS-Modulating Drugs 

Participants’ medication lists were collected from the MS clinic neurologist reports and prescribed medications were separated into two major groups of CNS-modulating drugs: (1) excitatory (e.g., dopamine agonists, serotonin receptor agonists or reuptake inhibitors, noradrenergic receptor agonists or reuptake inhibitors, levothyroxine, oral contraceptives, acetaminophen) and (2) inhibitory (e.g., beta-adrenergic blockers, anticholinergics, dopamine receptor antagonists, anticonvulsants, GABA-ergic medications) [11,111,112,113,114,115,116,117,118]. To compare these categorical variables across EDSS disability groups (EDSS ≥ 3 and <3), Pearson’s chi-square (χ2) test was used (Portney & Watkins, 2009), with Fisher’s exact test. Proportions of the following variables were compared across groups using Fisher’s exact test: (1) disease-modifying drugs (yes or no), (2) CNS inhibitory drugs (yes or no), (3) CNS excitatory drugs (yes or no), and (4) recreational drugs (yes or no).

## 3. Results

### 3.1. Participants

We recruited 223 people with MS, and 192 people consented to participate in the study (Figure 6). The final sample included 110 people with EDSS scores ranging from 0 to 7. Participant demographics are reported in Table 1. The total TMS assessment time ranged from 30–45 min per participant (mean ± SD: 37.75 ± 5.23).

### 3.2. Excitability Differences between Hemispheres

There were statistically significant differences in CSE (i.e., motor thresholds, eREC) between brain hemispheres (Table 2); however, these differences were largely driven by effects observed in the group with a higher level of disability (Table 2;Figure 7). When analyzing all participants, the hemisphere corresponding to the weaker hand demonstrated significantly lower CSE (i.e., higher AMT) compared to that of the stronger hand (z = 2.33, *p* = 0.020; Figure 7A). The participants with a greater level of disability had significantly lower CSE (i.e., higher AMT) in the weaker compared to the stronger hand (z = −3.20, *p* < 0.001; Figure 7B). However, in the group with a lower level of disability, no statistically significant differences between hemispheres for AMT were noted (z = −0.240, *p* = 0.811; Figure 7C). Similarly, when analyzing eREC parameters in the entire sample, the weaker side demonstrated significantly lower recruitment gain (i.e., lower slope; z = −2.10, *p* = 0.036) and lower overall excitation (i.e., lower AUC; t_(76)_ = 2.13, *p* = 0.037) compared to the stronger hand (Figure 8A,D). When analyzing groups separately, this statistically significant difference existed only in the group with more disability (slope: z = −3.29, *p* = 0.001; AUC: t_(17)_ = 2.56, *p* = 0.021; Figure 8B,E) and was not significantly different in the group with less disability (slope: z = −0.913, *p* = 0.361; AUC: t_(58)_ = 1.08, *p* = 0.286; Figure 8C,F).

Analysis of MEP amplitudes across all stimulated intensities (105–155% of AMT) revealed that, in all participants, MEP amplitudes were significantly lower in the weaker compared to the stronger hand at the middle to higher stimulation intensities (135–155% of AMT; t_(77)_ ≥ 2.18, *p* ≤ 0.032; Figure 8A). The participants with higher level of disability demonstrated significantly lower MEP amplitude in the weaker compared to the stronger hand at these intensities (135–155% of AMT; z = −1.73, *p* = 0.084, z = −3.06, *p* = 0.002, z = −2.72, *p* = 0.006, respectively; Figure 8B). In participants with lower level of disability, no statistically significant side-to-side differences were noted for MEP amplitudes at any of the stimulated intensities (z ≤ 0.18, *p* ≥ 0.854, t ≤ 1.20, *p* ≥ 0.234; Figure 8C). The accuracy (R^2^) of the eREC was significantly lower in the weaker compared to the stronger hand with all participants, as well as in participants with both more and less level of disability (z = −2.93, *p* = 0.003, z = −2.07, *p* = 0.039, t_(58)_ = 2.64, *p* = 0.011, respectively; Figure 8D–F).

### 3.3. Inhibitory Differences between Hemispheres (iREC)

When considering the entire participant sample, the hemisphere corresponding to the weaker hand demonstrated significantly longer CSP durations across all tested stimulation intensities compared to the stronger hand (105–155% of AMT; z ≥ 4.35, *p* ≤ 0.001, t ≥ 2.75, *p* ≤ 0.028; Figure 9A). When analyzing groups separately based on level of disability, the group with a greater level of disability demonstrated significantly longer CSP duration in the weaker compared to the stronger hand at the intensities of 105–145% of AMT (z ≥ 2.90, *p* ≤ 0.004, t ≥ 2.44, *p* ≤ 0.026; Figure 9B), whereas participants in the group with a lower level of disability demonstrated significantly longer CSP duration at lowest intensities (105–125% of AMT; z ≥ 2.38, *p* ≤ 0.017, t ≥ 2.62, *p* ≤ 0.011; Figure 9C). Significantly greater total inhibition (i.e., iREC AUC) was noted in the weaker compared to the stronger hand in all participants (z = 3.40, *p* = 0.001) as well as in the groups with more (t_(17)_ = 2.59, *p* = 0.019) and less disability (t_(58)_ = −2.82, *p* = 0.012), separately.

### 3.4. MEP Latencies Not Significantly Different between Hemispheres

When analyzing all participants, as well as separate groups based on level of disability, MEP latency did not significantly differ between weaker and stronger hands in either resting (MEPs obtained during RMT) or active (MEPs obtained during AMT) assessments (resting latency: z ≤ 1.54, *p* ≥ 0.124; active latency: z ≤ 1.79, *p* ≥ 0.074).

### 3.5. Transcallosal Inhibition Differences between Hemispheres

When considering the entire sample, the onset latency of the iSP was significantly delayed when measured in the weaker compared to the stronger hand (t_(53)_ = 2.06, *p* = 0.044). This statistically significant difference dissipated when participants were divided into groups based on disability levels (higher level of disability: t_(14)_ = 1.57, *p* = 0.139, lower level of disability: t_(38)_ = 1.38, *p* = 0.176), suggesting these separate analyses were likely underpowered. In the entire sample, there was no significant difference in iSP duration between hands (t_(53)_ = −1.72, *p* = 0.091). When separating the sample based on level of disability, the duration of the iSP was significantly longer in the stronger compared to the weaker hand in the group with a higher level of disability (z = 2.23, *p* = 0.026) but not the group with lower level of disability (t_(38)_ = −0.37, *p* = 0.713). In the entire sample, as well as with participants divided by disability levels, no other significant side-to-side differences were noted for any of the other iSP variables of (AUC, depth, iSP depth × duration; t ≤ 1.58, *p* ≥ 0.137).

### 3.6. Relationships between Clinical Outcomes and TMS Variables

Results of correlations between TMS variables and clinical (motor and non-motor) outcomes are detailed in Table 3. In general, there were consistent and statistically significant correlations between almost all TMS variables and *motor* outcomes (walking speed and 9HPT). There were also statistically significant correlations between some TMS variables and cognition (SDMT). There were far fewer statistically significant correlations with fatigue (AMT asymmetry and CSP) (Table 3). We noted that the hemisphere corresponding to the weaker side demonstrated stronger and more consistent correlations with 9HPT bilaterally (upper extremity dexterity), as well as SDMT (cognition).

### 3.7. Best Predictors of Walking Speed

The variables age and sex, included in the first block, did not contribute significantly to the variance in walking speed (R^2^ = 0.035, *p* = 0.183). After entering the TMS variables with the strongest relationships to walking speed, six TMS variables significantly explained variance in walking speed (Figure 10; R^2^ = 0.375, *p* < 0.001). Delayed onset of iSP was the strongest predictor of slower walking speed. 

### 3.8. Best Predictors of Upper Extremity Dexterity (9HPT)

The variables age and sex contributed significantly to 9HPT performance in both the stronger (R^2^ = 0.082, *p* = 0.047) and weaker hands (R^2^ = 0.093, *p* = 0.031). After entering the TMS variables with the strongest relationships to 9HPT, three TMS variables significantly explained variance in 9HPT performance in the stronger hand (Figure 10; R^2^ = 0.304, *p* < 0.001) and five TMS variables in the weaker hand (R^2^ = 0.353, *p* < 0.001). Interestingly, all the TMS variables which most significantly explained the variance in 9HPT performance were derived from the hemisphere corresponding to the *weaker* hand, regardless of which side 9HPT performance was being evaluated from (i.e., stronger or weaker). Although the R^2^ values suggest they were moderate predictors, measures of transcallosal inhibition were the most consistent and robust predictors bilaterally (Figure 10).

### 3.9. Best Predictors of Fatigue

The variables age and sex, included in the first block, did not contribute significantly to the variance in fatigue (R^2^ = 0.002, *p* = 0.905). After entering all statistically significant predictors (CSP duration of the weaker hand at 155% of AMT and AMT asymmetry) these TMS variables explained 15.5% of variance in fatigue (R^2^ = 0.155, *p* = 0.019). Note that only longer CSP duration of the weaker hand was a weak but statistically significant predictor of greater fatigue (Figure 10).

### 3.10. Best Predictors of Cognition (SDMT)

The variables age and sex, included in the first block, contributed significantly to SDMT performance, accounting for 10.1% of variance in SDMT (R^2^ = 0.101, *p* = 0.026). Among all collected TMS variables, poorer cognitive processing speed performance was better explained by higher RMT and lower MEP amplitudes assessed at 155% of AMT, in the hemisphere corresponding to the weaker hand only (R^2^ = 0.289, *p* = 0.039; Figure 10).

### 3.11. Use of CSN-Modulating Drugs

The two groups (i.e., EDSS ≥ 3 vs. <3) were not significantly different in terms of proportions of persons prescribed CNS excitatory drugs (χ^2^_(1)_ = 2.46, *p* = 0.141) or recreational drugs (χ^2^_(1)_ = 1.30, *p* = 0.292). A significantly greater proportion of participants in the high disability group (62%) were prescribed CNS inhibitory drugs than those in the low disability group ((χ^2^_(1)_ = 6.58, *p* = 0.013; 36%) (*p* < 0.05).

### 3.12. The Core-Set

Considering the correlations (Table 3) as well as their multicollinearity, and the regressions performed (Figure 10), the following are key elements of a core-set single-pulse TMS protocol to assess corticospinal excitability in clinical populations such as MS:Studies should consider assessing TMS bilaterally in order to index brain excitability asymmetries. However, the hemisphere corresponding to the weaker, or the more affected body side, should be prioritized.When investigating motor outcomes, studies should prioritize:a.Biomarkers of contra- and ipsilateral conduction latency.b.MEP amplitudes assessed at suprathreshold mid-high range TMS intensities (e.g., ≥125% to ≤145% of motor threshold).c.CSP assessed at lower-mid range TMS intensities (e.g., >100% to ≤115% of motor threshold).When investigating symptoms of fatigue, studies should prioritize:
CSP assessed at higher range TMS intensities (e.g., ≥145% of motor threshold).
When investigating cognition, studies should prioritize:a.RMT.b.MEP amplitudes assessed at suprathreshold mid-high range TMS intensities (e.g., ≥135% to 155% of motor threshold). 

## 4. Discussion

The integrity of the central nervous system, as well as progression of disease activity, is often probed using MRI [8,9]. However, accumulating evidence suggests inflammatory central nervous system lesions are transient on brain imaging [9] and symptoms relate more to disruption in functional brain networks that do not always localize well to individual brain structures [119]. As opposed to localizing structural foci of MS features, as is common in structural MRI, tools such as TMS can measure connectivity in real time [10]. Consequently, TMS could provide earlier evidence of central nervous system dysfunction, prior to the emergence of structural lesions on brain imaging [27]. The present study aimed to test and describe in detail the methodology of a single pulse TMS protocol that we performed in a large cohort of people with MS. We report three main findings. First, in terms of whether it is necessary to collect TMS variables bilaterally, the current findings indicate that there are significant differences in CSE, corticospinal inhibition, and transcallosal inhibition that depend on differentiating participants’ stronger and weaker hands. Indeed, the hemisphere corresponding to the weaker hand provided the most consistent and strongest predictors of clinical outcomes, suggesting a shift towards asymmetry could signal a more degenerative phase of MS [58]. Additionally, given the relationships we found between clinical outcomes and TMS variables from the hemisphere corresponding to the weaker hand, in a setting where there are time and logistical constraints, completing TMS testing in the weaker side, at least, should be top priority. Secondly, we showed that variables derived using TMS more strongly correlated with motor (walking speed and 9HPT) than non-motor outcomes (fatigue and cognition). However, some unique TMS variables significantly predicted fatigue (higher intensity-elicited CSP, suggestive of GABA_B_-receptor activity; Figure 10) and cognition (RMT as an index of global CSE and MEP amplitudes) suggesting they may have utility as biomarkers (albeit weak) for non-motor outcomes. Lastly, of the various TMS variables collected, five were good predictors of motor outcomes and two were weaker predictors of non-motor outcomes. Their predictive value was strengthened (R^2^) when used together. Motor thresholds (RMT, AMT), which are variables frequently reported across the literature [27], were not consistently predictive. The motor thresholds are the basic requirement to establish a MEP, but it was the characteristics of the MEP, such as its latency and the silent period, that were most strongly associated with clinical outcomes and should therefore be considered as part of a ‘core set’. Importantly, we found that the iSP, an indicator of transcallosal inhibition, was the strongest predictor of motor performance.

### 4.1. Asymmetry as a TMS Biomarker

We previously showed that people with MS displayed an asymmetric pattern of CSE (AMT asymmetry), in which higher excitability in the hemisphere corresponding to the weaker hand predicted earlier and less symptomatic MS, whereas a shift towards less excitability in the weaker hand predicted later and more symptomatic MS [58]. In MS, hyperexcitability is mediated early on by excitotoxicity and known to precede later neurodegeneration [36,37,38,120]; this could explain the gradual shift from higher to lower excitability in the more affected hemisphere as MS progresses. However, in the present work, no interhemispheric asymmetry of excitability was observed in MS participants with low levels of disability (EDSS < 3) but was evident in people with a high level of disability (EDSS ≥ 3). This suggests that individuals later in the natural course of MS with higher EDSS resembled people with stroke, in which there was an enduring link between hypoexcitability in the hemisphere corresponding to the weaker hand [30,121]. Interestingly, when we entered asymmetry as a separate TMS predictor in our modelling, it did not stand up to scrutiny; other variables were stronger (Figure 10). Nevertheless, the results we observed here and previously [58] suggest that AMT asymmetry may be a sensitive marker of disease progression. Its’ statistically significant correlation with TMS variables across all outcomes studied makes AMT asymmetry clinically robust. Consequently, we recommend gathering TMS data from both cerebral hemispheres routinely. We advise against collapsing data bilaterally across hemispheres (or defining hemispheres based on left vs. right or the traditionally dominant vs. non-dominant hand) because this approach may undermine the sensitivity inherent in indexing asymmetry based on clinical weakness. In the future, to better establish the validity of AMT asymmetry as a biomarker in MS, larger longitudinal studies should be performed across the natural course of MS to better understand whether the shift of asymmetry relates to hyperexcitability, mediated by excitotoxicity, and/or neurodegeneration, secondary to chronic neuroinflammation.

### 4.2. Robustness of TMS Variables from Hemisphere Corresponding to Weaker Hand

When we examined which TMS variables were the best predictors of clinical outcomes, measurements from the hemisphere corresponding to the weaker hand—namely motor thresholds, MEP amplitudes, and CSP duration—significantly accounted for the most variance in fatigue and cognition (SDMT). Moreover, hand dexterity (9HPT), regardless of which hand was examined, was strongly predicted by TMS variables derived from the weaker hand only (except for iSP). Taken together, these findings suggest there is a meaningful physiological difference between the stronger and weaker upper extremity motor pathways, which reflects a clinically meaningful impact on fine motor function, fatigue, and cognition. 

Findings from stroke research have shown that brain damage results in interhemispheric neurophysiological changes, described by the ‘interhemispheric competition’ model [30]. It is thought that excessive inhibition of the less affected to the more affected cerebral hemisphere through the corpus callosum may impair motor recovery of the affected limb [30]. This effect may be more pronounced later in the disease [122,123,124]. Moreover, other work suggests that compensatory activity from ipsilateral brain structures such as the dorsal premotor area may account for greater levels of inhibition in the more affected hemisphere [125,126]. These findings suggest that central nervous system damage secondary to chronic neurodegeneration may result in hypoexcitability and excess inhibition in the hemisphere corresponding to the more severely impaired upper extremity. Thus, TMS variables from the hemisphere corresponding to the weaker hand are a potentially important biomarker that should be, at minimum, included in the core set.

### 4.3. Significance of Transcallosal Inhibition and Elements of the iSP

In addition to comparing corticospinal excitability and inhibition across cerebral hemispheres, interhemispheric competition can also be indexed more directly using the iSP, a proxy for transcallosal inhibition [25,28]. In the current study, we found that iSP onset was significantly later in the hemisphere corresponding to the weaker hand, across the entire sample (Table 2). Yet, when participants were stratified based on low and high levels of disability, only the group with a higher level of disability was noted to have interhemispheric differences in iSP; the iSP duration was significantly longer in the stronger side. Additionally, iSP variables, particularly iSP onset latency, were some of the strongest predictors of both gross (walking speed) and fine (9HPT) motor function (Table 3). Previous work has shown that characteristics of the iSP (i.e., onset latency, duration, transcallosal conduction time) in people with MS reflect greater levels of transcallosal inhibition compared to healthy controls [31,82]. Others have shown that the degree of transcallosal inhibition is significantly related to level of disability based on EDSS [32,95], and that people with a low level of MS-related disability do not have abnormalities in transcallosal conduction time [122]. Importantly, transcallosal and interhemispheric inhibition are linked to microstructural integrity and lesion burden of the corpus callosum [31,82,127]. Indeed, our current findings show that there were no interhemispheric differences in iSP in people with a low level of disability. Future research should compare iSP among people with mild MS and healthy controls to determine whether the variable is sensitive enough to detect subtle interhemispheric changes early in the disease and whether these changes are amenable to treatment.

### 4.4. Single Pulse TMS to Investigate Potential for Neuroplasticity

In the context of clinical research and rehabilitation, the term ‘neuroplasticity’ is generally used to refer to positive excitatory changes that result from LTP, a mechanism characterized by the formation and strengthening of new neuronal connections [128,129,130,131,132]. Upregulation of glutamate receptors increases brain excitability and strengthens neuronal connections, whereas increased sensitivity to GABA does the opposite [131,133,134]. A highly excitable and disinhibited brain requires less intense TMS stimuli to evoke MEPs (i.e., lower motor thresholds), demonstrates higher excitatory recruitment curve characteristics (i.e., steeper gain properties, less variability, and greater total excitability), and exhibits shortened CSP duration and total inhibition. On the contrary, increased motor thresholds, smaller eREC values, and excessive intracortical inhibition mediated by both GABA_A_- and GABA_B_-receptor activity (i.e., long CSP duration) are all biomarkers of pathologically reduced CSE [135], brain damage (e.g., stroke, neurodegeneration) [136], and diminished potential for LTP [131,137]. Our results lend support that there is reduced capacity for LTP and neuroplasticity in people having higher disability due to MS. Mapping these CSE changes within an individual over time, after relapse, or as a result of a treatment, is an area worth further study.

### 4.5. How TMS Probes GABAergic-Mediated Corticospinal Inhibition

We used the single pulse CSP paradigm as a method to describe GABA_A_- and GABA_B_-receptor-mediated corticospinal inhibition, although this method has been disputed elsewhere [24]. Other approaches utilize paired pulse TMS paradigms to explore GABAergic inhibition [10]. Intracortical mechanisms can be studied such as SICI, LICI, and intracortical facilitation, which reportedly examine mechanisms related to GABA_A_, GABA_B_, and glutamate, respectively [11,27]. Paired pulse experiments require testing a wide range of stimulus intensities and interstimulus intervals due to the high inter- and intra-participant variability [138]. This comes with the expense of a lengthy experiment that is not always practical in clinical settings, and may also present difficulties in participants who have a high degree of spasticity and fatigability. Collecting CSPs at a wide range of stimulation intensities has been proposed as an alternative method to measure short- and long-lasting inhibition (GABA_A_- and GABA_B_-receptor activity, respectively) [10,135]. This procedure is relatively rapid, well-tolerated among participants, and allows for concurrent measurement of CSE, making it an ecologically valid approach to data collection in clinical populations like MS. 

In all participants, irrespective of level of disability, the differences between weaker and stronger hands was more obvious with shorter-lasting CSPs (reflecting GABA_A_ergic activity [10,135]), which were elicited at lower stimulated intensities (105–125% of AMT; Table 2). The level of statistical significance tended to gradually decrease towards the higher stimulation intensities (135–155% of AMT). It is possible that this trend favouring CSPs that index GABA_A_ activity reflects a role for GABA_A_, more so than GABA_B_, as a better biomarker of disease progression during the earlier stages of MS. There is research in MS showing that SICI changes (which reflect GABA_A_ activity [10,135]) differentiate people with MS from controls and relate to MS symptoms and disease progression [123,124,139]. Presently, it was interesting to note that the CSPs, believed to be more reflective of GABA_B_-receptor activity (elicited at higher stimulation intensities), in the hemisphere corresponding to the weaker hand were correlated with fatigue, but not those reflective of GABA_A_ (Table 3). Therefore, it is possible that the level of GABA_A_ versus GABA_B_ inhibitory activity may be indicative of different deficits (e.g., motor vs. non-motor). Such information is important to consider when developing targeted drug and rehabilitation interventions to promote neuroplasticity and restore function.

### 4.6. Using the Recruitment Curve to Examine Glutamatergic-Mediated Corticospinal Excitation

When analyzing each stimulated intensity on the recruitment curve (105–155% of AMT), it was noticeable that lower simulator intensities (105–125% of AMT) showed greater interhemispheric differences in MEP amplitude between weaker and stronger sides (Table 2). Classically, the stimulus-response relationship (stimulus intensity × MEP amplitude) of the eREC is described by a sigmoidal function [10,13], whereby the curve starts as a flat line at sub-threshold TMS intensities and increases in a linear fashion as a function of increasing stimulus intensity, until a plateau is reached where there is no further increase in MEP amplitude despite a further increase in stimulus intensity [14,15,16]. The ascending portion (which we measured using the current protocol of 105–155% of AMT) is thought to represent the activity of glutamatergic neurons [14,140] with an increasing excitability threshold [141], while the plateau is indicative of increasing phase cancellation of motor unit action potentials that contribute to the MEP [142]. Past work has found a significant difference in MEP recruitment curve slope between people with RRMS and healthy controls, a negative relationship between the eREC slope and EDSS score, and helped predict variance in disability level to a greater degree than clinical characteristics such as age, disease duration, and sex [32]. Unlike the present findings, these authors attributed changes in eREC slope in MS to elements of the recruitment curve related to higher stimulation intensities, such as neurons spatially further away from the TMS hotspot, neurons that are intrinsically less excitable, or neurons with a greater degree of dysfunction due to MS-related cortical damage or demyelination [32]. Conversely, our work suggests that lower excitability of the intrinsically more excitable neurons, those closer to the motor hotspot, or those with a lower level of MS-related dysfunction seem to differ across the more vs. less severely affected hemisphere. This was also reflected by our findings related to AMT and AMT asymmetry. Therefore, further work will be necessary to elucidate which elements of the eREC are impacted by MS and at which disease stage and type or level of disability.

### 4.7. TMS Variables That Did Not Stand Up to Scrutiny

Two key variables, RMT and MEP latency, which are frequently reported in the literature, did not consistently relate to clinical outcomes and were overshadowed by other, more sensitive TMS biomarkers. A typical TMS assessment starts with the assessment of motor thresholds (most typically RMT) [10]. Indeed, motor thresholds are among the most commonly reported TMS variables in the MS literature [27]. We found that AMT served as a better biomarker than RMT. RMT is collected during complete muscle relaxation, and because in this condition, corticospinal motor neurons are below firing threshold and MEPs at RMT likely result from the summation of many I-waves from cortico-cortical connections [10,11]. Physiologically, the difference between RMT and AMT is not entirely known [10,11]. However, when compared to RMT, MEPs are more easily elicited during AMT assessment (requiring lower stimulation intensity), which implies that previously recruited I-waves from the individual’s own voluntary motor drive (i.e., already-firing motor neurons) brings motor neurons closer to their firing threshold. In light of this effect from voluntary motor drive, AMT-evoked MEPs likely result more from D-waves (from direct stimulation of the corticospinal tract) than I-waves (which rely on interneuronal populations) and may evaluate more directly the axonal threshold (rather than temporo-spatial summation) and deeper corticospinal tract neurons [10,11]. Although collecting both RMT and AMT is often considered standard practice [13], many MS studies have investigated RMT while relatively few have investigated AMT [27]. Our findings suggest that studies should consider collecting AMT preferably over RMT when studying CSE in MS. 

Throughout the MS literature, MEP latencies are among the most frequently reported and clinically significant TMS variables [27]. Based on the extant literature, a statistically significant difference between weaker and stronger hands in MEP latency was expected. However, we did not observe any statistically significant differences between hemispheres in terms of MEP latency. In the participants with lower levels of disability, there was a trend towards a statistically significant interhemispheric difference whereby MEP latencies tended to be longer in the hemisphere corresponding to the weaker hand. Nevertheless, when considered across the entire sample, this trend was effectively washed out by including participants with higher levels of disability. Therefore, we interpret this to mean that interhemispheric asymmetry in MEP latencies may demonstrate earlier demyelination. However, past work has shown the opposite;that people with progressive MS [123,124,139,143] and those with more clinical impairment [123,124,139] tend to have longer MEP latencies, while MEP latency also has a positive relationship with fatigue [143] and disease severity measured by EDSS [143,144,145]. Thus, other evidence suggests a putative deleterious effect of demyelination on MEP latency. The present study may indicate that early disease effects on MEP latency may be more related to desynchronization and phase cancellation in corticospinal neuron firing, possibly in connection with excitotoxicity [27,120,146]. Future work should examine the role of MEP latency to characterize differences between early and late disease phenotypes. Furthermore. MEP latency is influenced by muscle activity (resting vs. active). MEPs derived from contracting muscle have shorter MEP latencies (~2 ms faster) for reasons that are yet to be determined [10]. Since MEPs derived from contracted and resting muscles are likely governed by different brain structures and processes [147,148], future research should attempt to understand MEP latency differences in clinical populations.

### 4.8. Limitations

Although we present TMS data from the largest cohort of people with MS collected to date, there are some limitations. We did not examine CSE in age and sex-matched controls, nor did we examine the change in TMS variables longitudinally. Future studies should attempt to fill these gaps. Additionally, we did not compare results obtained using paired pulse TMS paradigms to those from our singe-pulse protocol. This limits our ability to provide a head-to-head comparison on these differing TMS protocols. It is important to appreciate that drugs that modulate the CNS likely influence CSE. We report that in this cohort, inhibitory (but not excitatory) modulating CNS drugs were more frequently prescribed among participants with higher levels of disability. Changes in TMS biomarkers of cortical inhibition (e.g., CSP) in MS could be related, in part, to medications (e.g., for management of chronic pain, spasticity, sleep issues, anxiety, continence issues). Research that focus primarily on elucidating the effects of drugs on the CSE of people with MS is needed.

## 5. Conclusions

We aimed to clearly and comprehensively describe the methodology of a single pulse TMS protocol. Using data from 110 people with MS, we showed how the variables derived could be used to probe central nervous system (dys)function. Specifically, we found that delayed and longer iSP (a measure of transcallosal inhibition; the influence of one brain hemisphere’s activity over the other) consistently predicted slower walking speed and poorer hand function. Longer cortical silent period (suggestive of greater corticospinal inhibition via GABA) was the most robust predictor of fatigue while higher RMT (lower corticospinal excitability) was the best predictor of slower cognitive processing speed. Greater interhemispheric asymmetry (imbalance between hemispheres) of participants’ corticospinal excitability (measured using AMT) was significantly correlated with overall poorer performance in the greatest number of clinical outcomes. Indeed, although this asymmetry was not detectable in people with milder MS (EDSS < 2.5), the differences in excitability between hemispheres became more apparent once EDSS reached 3.0. Notably, values derived from the hemisphere corresponding to the weaker hand resulted in the strongest relationships to clinical (motor and non-motor) outcomes, suggesting that, as a minimum, measurements should be taken from this side. We also show that TMS variables related more strongly to motor outcomes than non-motor outcomes. Our findings support the idea that TMS should be considered a potential biomarker to identify, characterize, and monitor MS based on the stage of excitotoxic vs. neurodegenerative disease and clinical disability. We have provided a simple methodological pipeline to examine excitatory and inhibitory corticospinal mechanisms in MS that map to clinical status. The work outlined here is a starting point to better grow the body of TMS work in MS, using a clinically relevant and logistically friendly core set of TMS protocols that can be expanded across testing sites and better evaluated in larger prospective and longitudinal studies.

## Figures and Tables

**Figure 1 brainsci-11-00384-f001:**
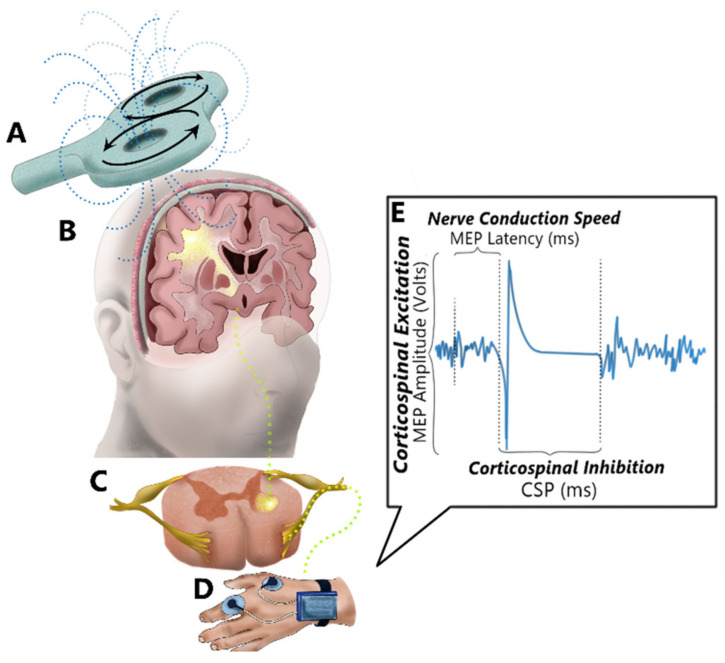
Basic Neurophysiological Principles of Transcranial Magnetic Stimulation (TMS). Electrical current produced by the stimulator travels via an insulated wire and reaches the stimulator coil (e.g., figure of eight coil) (**A**). The direction of flow of the electrical current within the coil (black arrows) is able to generate a perpendicular magnetic field (blue dotted lines), that (**B**) passes through the scalp painlessly and activates corticospinal neurons in the primary motor area by electromagnetic induction. (**C**) TMS elicits descending corticospinal volleys from the brain to the spinal cord by directly activating pyramidal tract neurons or indirectly via interneurons that synapse on the pyramidal tract (D- and I-waves, respectively); these signals elicit a motor evoked potential (MEP) in the contralateral muscle under investigation (e.g., first dorsal interosseous muscle). (**D**) TMS-induced MEPs are recorded via electromyography (EMG), with recording electrodes placed over the belly of the target muscle. (**E**) Offline analysis of corticospinal excitability (e.g., MEP peak-to-peak amplitude) and intracortical inhibition (e.g., cortical silent period (CSP) duration; MEP onset to return of EMG background activity), and corticomotoneuronal conduction speed (e.g., MEP latency; time from TMS stimulus to MEP onset) from a TMS-elicited MEP recorded by EMG of the first dorsal interosseous muscle with a participant performing a tonic voluntary contraction (e.g., pinch grip). Original figure © Arthur R. Chaves (created on Autodesk^®^ Sketchbook^®^ free software).

**Figure 2 brainsci-11-00384-f002:**
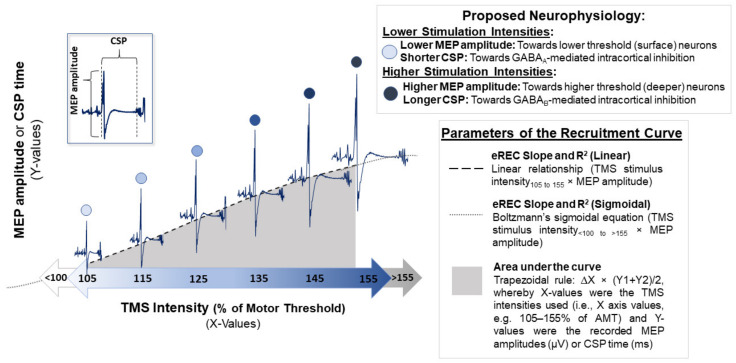
Parameters and Neurophysiology of the Excitatory and Inhibitory Recruitment Curves. Excitatory recruitment curves (eREC) are investigated by applying a varying range of transcranial magnetic stimulation (TMS) intensities [i.e., maximal stimulator output percentage (MSO%), e.g., 100–155% of Motor Threshold] and investigating the subsequent increases in motor evoked potential (MEP) amplitude (Volts). A linear or sigmoidal (Boltzmann’s) plot between MEP amplitude (y-values) versus the TMS intensities used (x-values) will determine the REC parameters [i.e., slope, and r-squared (R^2^)]. When muscle tone (tonic voluntary contraction) is performed by the participant during the REC assessment, the TMS variable cortical silent period (CSP), a biomarker of intracortical inhibition, can be investigated and the inhibitory REC (iREC) can be assessed. Overall excitation and inhibition are assessed by calculating the area under the curve using the trapezoid rule.

**Figure 3 brainsci-11-00384-f003:**
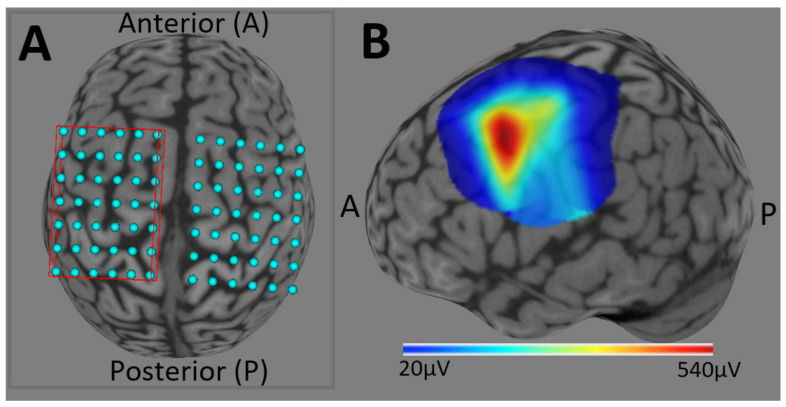
Finding the Hotspot and Creating a Motor Map. Guided using neuronavigation software, a 6 cm × 7 cm grid is placed on the motor area to assist with finding the first dorsal interosseous primary motor area representation corresponding to the right hand (**A**). Blue dots (12 mm apart) represent the pre-determined targets in which the experimenter performed 2–3 transcranial magnetic stimulation (TMS) stimulations. (**B**) A “heat map” is built using the neuronavigation software (Brainsight, Rogue Research Inc, Montreal, QC, Canada) demonstrating the primary motor area’s first dorsal interosseous muscle representation from a single multiple sclerosis (MS) participant (female, 31 years-old, presenting with no disability [expanded disability status scale (EDSS) 0].

**Figure 4 brainsci-11-00384-f004:**
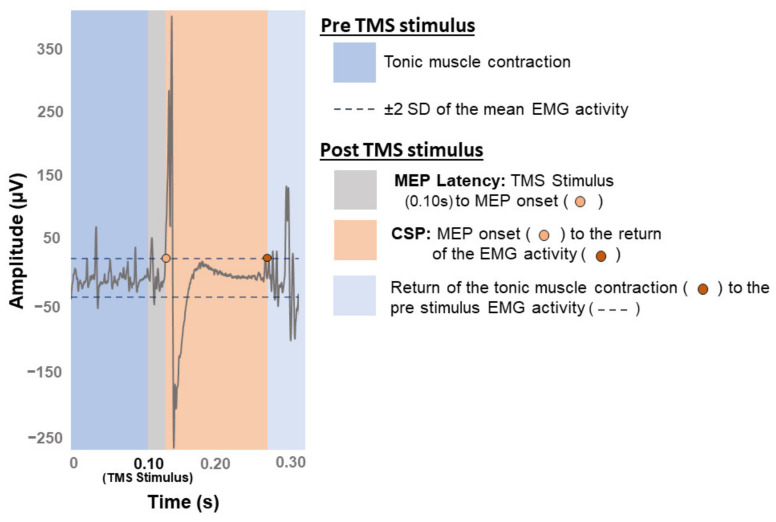
Deriving data from a contralateral Transcranial Magnetic Stimulation (TMS)-induced Motor Evoked Potential (MEP). Representative MEP from contralateral hand muscle showing background electromyography (EMG) activity (tonic contraction) before the TMS stimulus (0.10 s on the timescale). EMG change is evaluated based on ±2 SD from the mean indicated as dashed lines. MEP latency, peak-to-peak MEP amplitude, and length of the cortical silent period (CSP is derived during offline analysis.

**Figure 5 brainsci-11-00384-f005:**
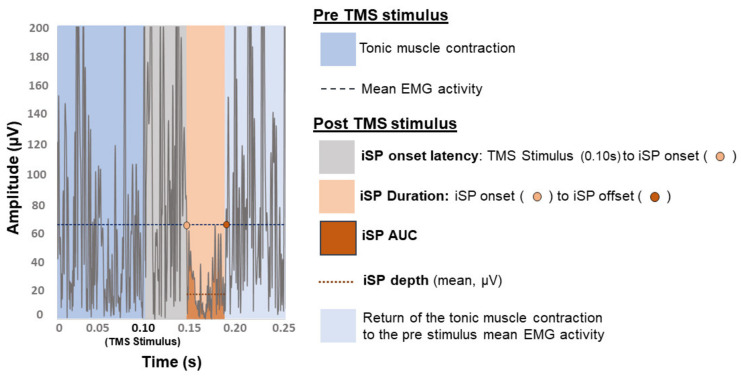
Deriving data from a Transcranial Magnetic Stimulation (TMS)-induced Ipsilateral Silent Period (iSP). Representative ipsilateral silent period (iSP) from ipsilateral hand muscle, during maximal pinch grip, showing electromyography (EMG) activity before the TMS stimulus (0.10 s on the timescale). EMG activity is briefly suppressed after TMS stimulus. During offline analysis, iSP onset latency, duration, depth, and area under the curve (AUC) can be calculated.

**Figure 6 brainsci-11-00384-f006:**
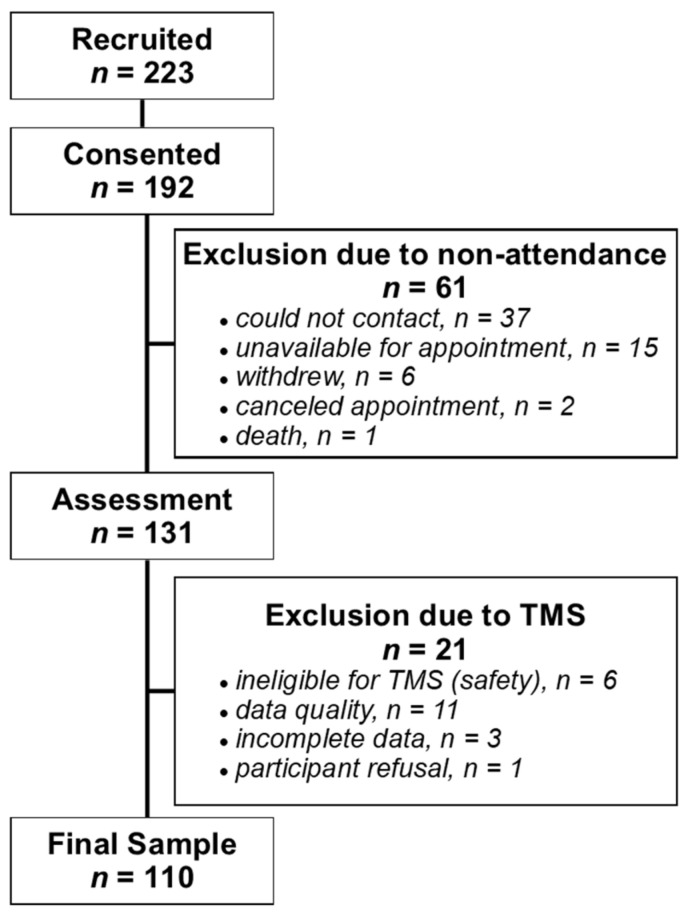
Recruitment of Participants.

**Figure 7 brainsci-11-00384-f007:**
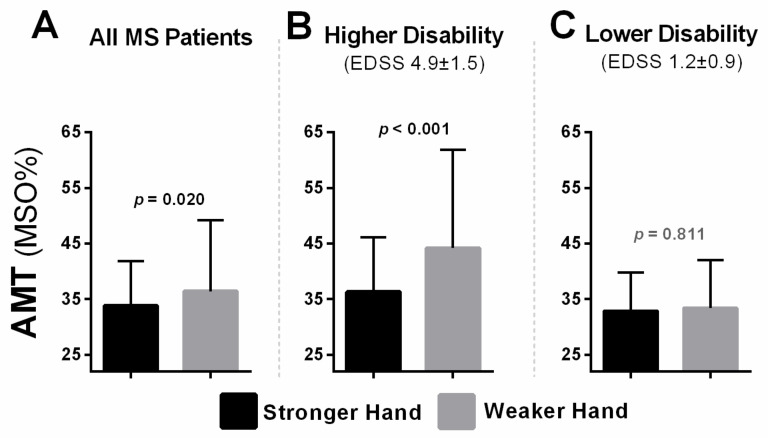
Corticospinal Excitability (CSE) Differences Between Hemispheres. (**A**) Considering all participants in the same analysis, significantly lower CSE was noted in the hemisphere corresponding to the weaker than to the stronger hand. When separating participants into groups of higher (EDSS ≥ 3) vs. lower (EDSS < 3) level of disability, (**B**) only the group with higher level of disability demonstrated significantly higher active motor threshold (AMT) (i.e., lower CSE) in the hemisphere corresponding to the weaker compared to the stronger hand. (**C**) No significant difference between hemispheres was noted for AMT in the group with lower level of disability. Error bars represent one standard deviation (±SD) of the mean. EDSS, expanded disability status scale (MS severity; 0 = no disability, to 10 = death due to MS).

**Figure 8 brainsci-11-00384-f008:**
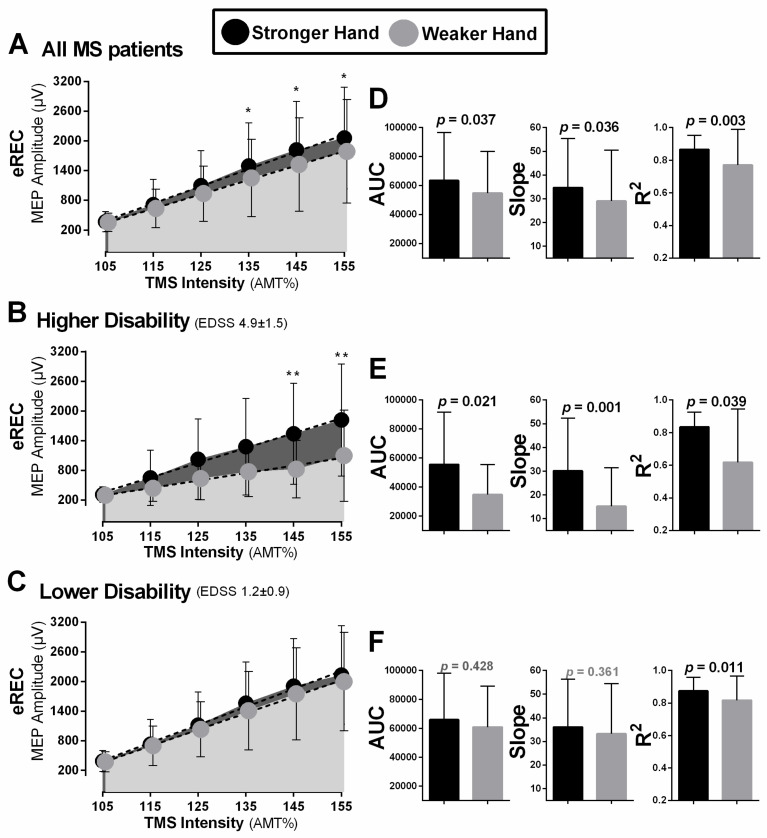
Differences in the Excitatory Recruitment Curve (eREC). (**A**) With all participants in the same analysis, when compared to the hemisphere corresponding to the stronger hand, the hemisphere corresponding to the weaker hand demonstrated significantly lower motor evoked potential (MEP) amplitudes (µV) at the transcranial magnetic stimulation (TMS) intensities of 135–155% of the AMT (*, *p* < 0.05). When groups were stratified based on levels of disability, (**B**) only the group with higher level of disability (EDSS) ≥ 3) demonstrated lower MEP amplitudes in the weaker hand (**, *p* < 0.01) at the intensities of 145 and 155% of the active motor threshold (AMT), whereas (**C**) no statistically significant difference between hemispheres was noted across MEP amplitudes (105–155% of AMT) in the group with lower level of disability (EDSS < 3). (**D**) With all participants in the same analysis, the eREC parameters of overall excitation (AUC, area under the curve), gain (slope), and accuracy (R^2^) were significantly lower in the weaker compared to the stronger hand. When separating participants into groups based on disability levels, (**E**) the group with a higher level of disability (EDSS ≥ 3) demonstrated significantly lower excitatory recruitment curve gain (slope), overall excitation (AUC), and accuracy (R^2^) in the hemisphere corresponding to the weaker hand when compared to the stronger hand. (**F**) In the group with a lower level of disability, overall excitation (AUC) and gain (slope) did not significantly differ between hemispheres, whereas accuracy (R^2^) was significantly lower in the hemisphere corresponding to the weaker hand. Error bars represent one standard deviation (±SD) of the mean. EDSS, expanded disability status scale (MS severity; 0 = no disability, to 10 = death due to MS).

**Figure 9 brainsci-11-00384-f009:**
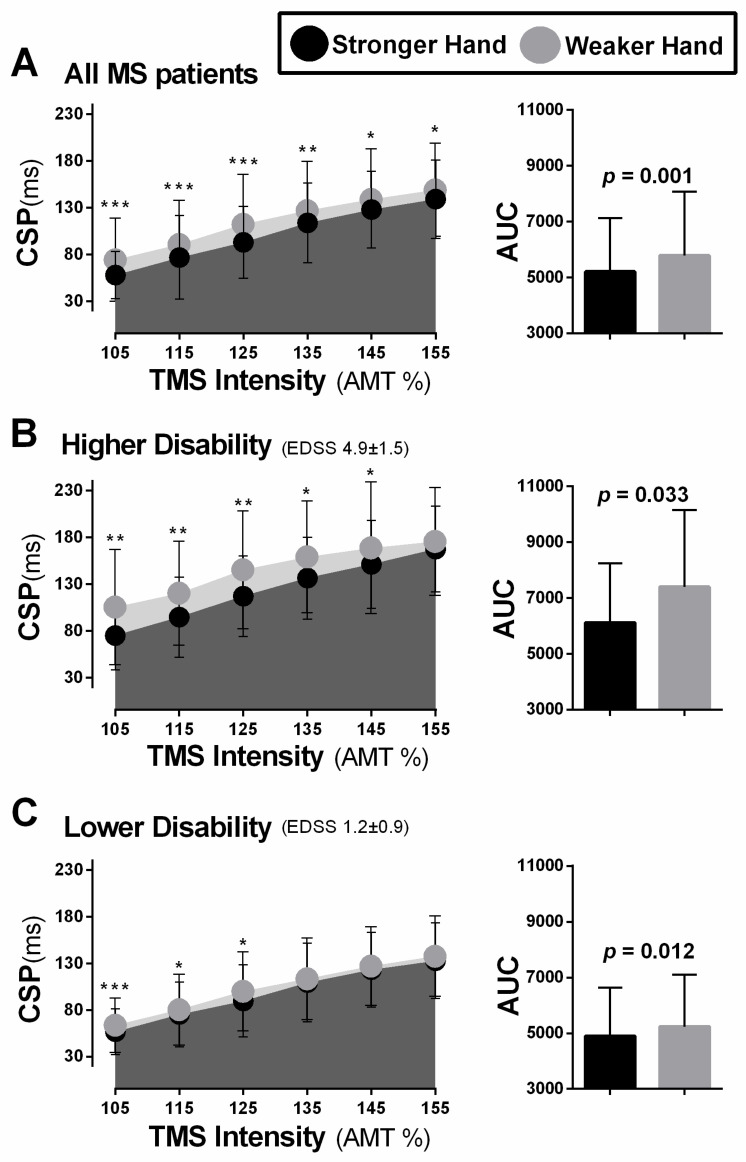
Differences in the Inhibitory Recruitment Curve (iREC). (**A**) With all participants in the same analysis, significantly longer cortical silent period (CSP) duration was noted in the hemisphere corresponding to the weaker hand compared to the stronger hand across all the iREC intensities. (**B**) In participants with a higher level of disability (EDSS) ≥ 3), CSP duration was significantly longer in the weaker compared to the stronger hand at TMS intensities of 105–145% of the active motor threshold (AMT). (**C**) In participants with a lower level of disability (EDSS < 3), CSP duration was significantly longer in the weaker compared to the stronger hand at the TMS intensities of 105–125% of AMT (**A**–**C**). In all participants, as well groups based on higher and lower level of disability, overall inhibition (AUC) was significantly higher in the weaker compared to the stronger hand. ***, *p* < 0.001, **, *p* < 0.010, *, *p* < 0.050. Error bars represent one standard deviation (±SD) of the data mean. EDSS, expanded disability status scale (MS severity; 0 = no disability, to 10 = death due to MS).

**Figure 10 brainsci-11-00384-f010:**
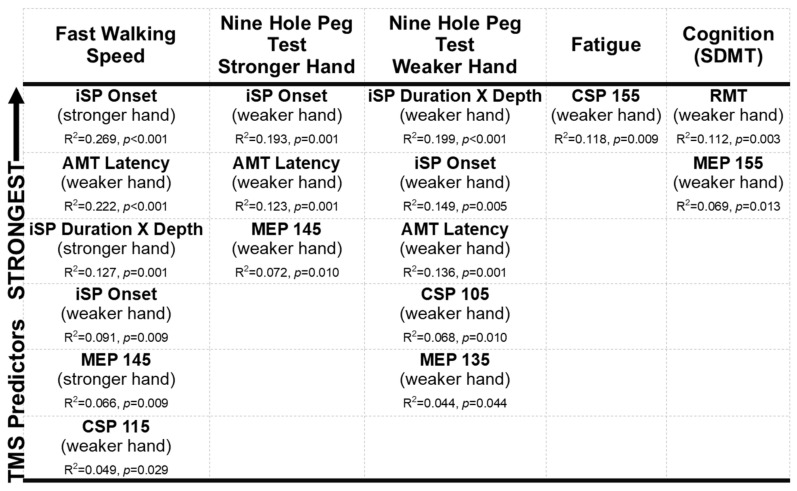
Transcranial magnetic stimulation (TMS) Variables that Most Strongly Predict Clinical Outcomes. AMT, active motor threshold; CSP, cortical silent period; iSP, ipsilateral silent period; MEP, motor evoked potential; RMT, resting motor threshold; SDMT, symbol digit modality test.

**Table 1 brainsci-11-00384-t001:** Participants’ characteristics.

		All Participants (*n* = 110)	Higher Levels of Disability Group (*n* = 34)	Lower Levels of Disability Group (*n* = 76)
Sex (Female/Male)		80/30	23/11	57/19
MS Type (n)	RRMS	91	16	75
	SPMS	14	13	1
	PPMS	5	5	-
Age (years)		48.4 ± 10.5	51.5 ± 11.3	47.0 ± 9.9
Disease Duration (years)		14.0 ± 8.4	17.18 ± 9.34	12.6 ± 7.5
MS Severity [EDSS 0–10; median (range)]		2.0 (0–7.0)	6.0 (3.0–7.0)	1.0 (0–2.5)
Functional Tests				
Walking Speed	cm/s	100.25 ± 28.73	74.36 ± 29.09	111.83 ± 19.68
	cm/s/height_cm_	0.59 ± 0.17	0.44 ± 0.17	0.66 ± 0.12
Upper Extremity [9HPT (seconds)]	Weaker	24.42 ± 8.13	30.20 ± 11.04	22.52 ± 5.87
	Stronger	22.27 ± 4.84	26.83 ± 6.17	20.71 ± 3.02
Fatigue [Low (0 mm) to High (100 mm)]		39.5 ± 30.1	37.8 ± 11.0	40.2 ± 31.4
Cognitive Processing Speed (SDMT score)		48.39 ± 10.82	41.21 ± 10.61	50.69 ± 9.90

Note: Data presented as mean ± SD, with exception of EDSS, presented as median (range). EDSS, Expanded Disability Status Scale; MS, Multiple Sclerosis; RRMS, relapsing remitting MS; SPMS, secondary progressive MS; PPMS, primary progressive MS; SDMT, symbol digit modality test; 9HPT, nine-hole peg test.

**Table 2 brainsci-11-00384-t002:** Differences in TMS variables between hemispheres.

	All Participants (*n* = 110)		Higher Level of Disability (EDSS 4.9 ± 1.5, *n* = 34)		Lower Level of Disability (EDSS 1.2 ± 0.9, *n* = 76)	
TMS Variables	Stronger Hand	Weaker Hand	Stronger Hand	Weaker Hand	Stronger Hand	Weaker Hand
**RMT (MSO%)**	41 ± 11 (*n* = 99)	43 ± 14 (*n* = 96)	44 ± 14 (*n* = 29)	50 ± 20 (*n* = 30)	39 ± 9 (*n* = 70)	40 ± 8 (*n* = 66)
**AMT (MSO%)**	34 ± 8 (*n* = 103)	37 ± 13 (*n* = 103)	36 ± 10 (*n* = 30)	44 ± 17 (*n* = 30)	33 ± 7 (*n* = 73)	34 ± 10 (*n* = 73)
**Resting MEP Latency** **(ms; ms/height_cm_)**	24.50 ± 3.70; 0.144 ± 0.022 (*n* = 96)	25.11 ± 4.11; 0.148 ± 0.023 (*n* = 92)	26.88 ± 5.63; 0.159 ± 0.34 (*n* = 29)	27.45 ± 5.75; 0.160 ± 0.33 (*n* = 29)	23.48 ± 1.6; 0.138 ± 0.01 (*n* = 67)	24.03 ± 2.5; 0.141 ± 0.13 (*n* =60)
**Active MEP Latency** **(ms; ms/height_cm_)**	23.60 ± 3.41; 0.139 ± 0.019 (*n* = 101)	24.20 ± 4.18; 0.143 ± 0.023 (*n* = 102)	26.14 ± 4.82; 0.154 ± 0.28 (*n* = 30)	26.89 ± 5.97; 0.158 ± 0.33 (*n* = 30)	22.52 ± 1.7; 0.133 ± 0.01 (*n* = 71)	23.02 ± 2.3; 0.136 ± 0.012 (*n* = 71)
**MEP Amplitude** **105% AMT (µV)**	366.47 ± 199.88 (*n* = 84)	352.49 ± 184.99 (*n* = 82)	313.95 ± 150.34 (*n* = 22)	297.97 ± 97.99 (*n* = 21)	385.11 ± 212.68 (*n* = 62)	371.26 ± 203.96 (*n* = 61)
**MEP Amplitude** **115% AMT (µV)**	695.69 ± 504.93 (*n* = 84)	626.97 ± 384.11 (*n* = 82)	611.07 ± 530.65 (*n* = 22)	437.40 ± 250.03 (*n* = 21)	725.71 ± 496.46 (*n* = 62)	692.23 ± 401.79 (*n* = 61)
**MEP Amplitude** **125% AMT (µV)**	1072.55 ± 697.87 (*n* = 84)	920.43 ± 554.91 (*n* = 82)	962.64 ± 788.99 (*n* = 22)	623.63 ± 408.55 (*n* = 21)	1111.55 ± 665.18 (*n* = 62)	1022.61 ± 564.46 (*n* = 61)
**MEP Amplitude** **135% AMT (µV)**	1442.97 ± 870.71 (*n* = 84)	1246.99 ± 773.34 (*n* = 80)	1204.49 ± 944.31 (*n* = 22)	787.22 ± 482.55 (*n* = 21)	1527.60 ± 834.74 (*n* = 62)	1410.63 ± 794.03 (*n* = 59)
**MEP Amplitude** **145% AMT (µV)**	1757.19 ± 985.46 (*n* = 84)	1507.81 ± 941.55 (*n* = 80)	1465.85 ± 1019.74 (*n* = 22)	819.45 ± 556.27 (*n* = 21)	1860.57 ± 960.16 (*n* = 62)	1752.82 ± 931.75 (*n* = 59)
**MEP Amplitude** **155% AMT (µV)**	1986.81 ± 1034.14 (*n* = 84)	1768.90 ± 1044.18 (*n* = 79)	1713.77 ± 1112.91 (*n* = 22)	1084.56 ± 883.46 (*n* = 20)	2083.70 ± 996.15 (*n* = 62)	2000.88 ± 997.18 (*n* = 59)
**eREC AUC**	61,450.39 ± 33,185.46 (*n* = 84)	54,323.52 ± 28,516.35 (*n* = 79)	52,579.10 ± 35,594.71 (*n* = 22)	34,558.14 ± 19,665.86 (*n* = 20)	64,598.27 ± 31,997.28 (*n* = 62)	61,023.65 ± 28,044.92 (*n* = 59)
**eREC Slope**	33.31 ± 20.69 (*n* = 84)	28.64 ± 21.35 (*n* = 79)	28.02 ± 21.76 (*n* = 22)	15.04 ± 15.69 (*n* = 20)	35.18 ± 20.14 (*n* = 62)	33.25 ± 21.14 (*n* = 59)
**eREC R** ^2^	0.86 ± 0.10 (*n* = 84)	0.77 ± 0.22 (*n* = 79)	0.80 ± 0.13 (*n* = 22)	0.61 ± 0.32 (*n* = 20)	0.88 ± 0.8 (*n* = 62)	0.82 ± 0.15 (*n* = 59)
**CSP Duration** **105% AMT (ms)**	61.34 ± 29.24 (*n* = 83)	74.62 ± 43.96 (*n* = 80)	75.13 ± 36.66 (*n* = 22)	105.09 ± 61.64 (*n* = 21)	56.37 ± 24.56 (*n* = 61)	63.78 ± 29.44 (*n* = 59)
**CSP Duration** **115% AMT (ms)**	80.46 ± 37.93 (*n* = 82)	90.97 ± 46.47 (*n* = 79)	94.71 ± 42.92 (*n* = 22)	119.98 ± 55.75 (*n* = 21)	75.24 ± 34.88 (*n* = 60)	80.46 ± 37.98 (*n* = 58)
**CSP Duration** **125% AMT (ms)**	97.16 ± 41.50 (*n* = 82)	112.16 ± 52.39 (*n* = 79)	117.24 ± 43.01 (*n* = 22)	145.07 ± 63.00 (*n* = 21)	89.80 ± 38.74 (*n* = 60)	100.25 ± 42.65 (*n* = 58)
**CSP Duration** **135% AMT (ms)**	116.64 ± 44.03 (*n* = 82)	126.03 ± 52.42 (*n* = 77)	136.43 ± 43.82 (*n* = 22)	158.89 ± 59.92 (*n* = 21)	109.39 ± 42.17 (*n* = 60)	113.71 ± 43.86 (*n* = 58)
**CSP Duration** **145% AMT (ms)**	130.82 ± 43.49 (*n* = 82)	138.34 ± 54.00 (*n* = 75)	151.15 ± 46.81 (*n* = 22)	168.51 ± 70.46 (*n* = 20)	123.37 ± 40.07 (*n* = 60)	127.36 ± 42.30 (*n* = 58)
**CSP Duration** **155% AMT (ms)**	142.20 ± 44.42 (*n* = 81)	148.45 ± 50.08 (*n* = 77)	167.50 ± 45.83 (*n* = 22)	175.39 ± 57.57 (*n* = 20)	132.92 ± 40.43 (*n* = 60)	139.00 ± 43.92 (*n* = 58)
**iREC AUC**	5278.49 ± 1881.28 (*n* = 81)	5738.80 ± 2266.99 (*n* = 75)	6208.30 ± 2028.57 (*n* = 22)	7150.18 ± 2727.68 (*n* = 20)	4931.78 ± 1715.08 (*n* = 59)	5225.58 ± 1848.99 (*n* = 55)
**iSP Onset Latency (ms)**	35.83 ± 7.70 (*n* = 55)	38.33 ± 9.19 (*n* = 55)	41.47 ± 9.25 (*n* = 15)	45.60 ± 10.13 (*n* = 15)	33.71 ± 5.88 (*n* = 40)	35.61 ± 7.21 (*n* = 40)
**iSP Duration (ms)**	30.10 ± 12.67 (*n* = 55)	26.37 ± 13.36 (*n* = 55)	36.42 ± 15.53 (*n* = 15)	25.74 ± 15.85 (*n* = 15)	27.73 ± 10.71 (*n* = 40)	26.61 ± 12.52 (*n* = 40)
**iSP AUC**	49.50 ± 12.48 (*n* = 55)	47.62 ± 14.81 (*n* = 55)	42.91 ± 11.31 (*n* = 15)	40.79 ± 18.66 (*n* = 15)	51.97 ± 12.11 (*n* = 40)	50.18 ± 12.41 (*n* = 40)
**iSP Depth**	50.77 ± 11.73 (*n* = 55)	53.46 ± 12.02 (*n* = 55)	55.60 ± 13.03 (*n* = 15)	58.11 ± 14.32 (*n* = 15)	48.97 ± 10.83 (*n* = 40)	51.71 ± 10.72 (*n* = 40)
**iSP Depth × Duration**	1484.66 ± 632.71 (*n* = 55)	1363.79 ± 723.69 (*n* = 55)	1897.03 ± 639.95 (*n* = 15)	1442.37 ± 940.06 (*n* = 15)	1330.03 ± 563.33 (*n* = 40)	1334.32 ± 636.13 (*n* = 40)
Statistically significant findings (sig.) are highlighted as:	Sig. ≤ 0.001	Sig. ≤ 0.010	Sig. ≤ 0.050			

Note: AMT, active motor threshold; CSP, cortical silent period; EDSS, expanded disability status scale; eREC, excitatory recruitment curve; iREC, inhibitory recruitment curve; iSP, ipsilateral silent period; MEP, Motor evoked potential; MSO, maximal stimulator output.

**Table 3 brainsci-11-00384-t003:** Identifying significant relationships (bivariate) between the transcranial magnetic stimulation variables and clinical outcomes (absolute^#^ r-value, and *p*-value).

	Walking Speed		9HPT Stronger Hand		9HPT Weaker Hand		Fatigue		SDMT	
TMS Variables	Stronger Hand	Weaker Hand	Stronger Hand	Weaker Hand	Stronger Hand	Weaker Hand	Stronger Hand	Weaker Hand	Stronger Hand	Weaker Hand
**RMT (MSO%)**	r = 0.138	r = 0.286	r = 0.247	r = 0.333	r = 0.212	r = 0.333	r = 0.048	r = 0.179	r = 0.167	r = 0.358
**AMT (MSO%)**	r = 0.211	r = 0.275	r = 0.146	r = 0.340	r = 0.203	r = 0.391	r = 0.076	r = 0.161	r = 0.095	r = 0.272
**RMT Asymmetry**	r = 0.214		r = 0.115		r = 0.183		r = 0.204		r = 0.253	
**AMT Asymmetry**	r = 0.274		r = 0.290		r = 0.329		r = 0.284		r = 0.311	
**Resting MEP Latency** **(ms; ms/height_cm_**)	r = 0.234	r = 0.314	r = 0.242	r = 0.463	r = 0.309	r = 0.422	r = 0.033	r = 0.092	r = 0.160	r = 0.217
**Active MEP Latency** **(ms; ms/height_cm_)**	r = 0.424	r = 0.504	r = 0.401	r = 0.421	r = 0.390	r = 0.449	r = 0.097	r = 0.050	r = 0.280	r = 0.342
**MEP Amplitude** **105% AMT (µV)**	r = 0.172	r = 0.146	r = 0.149	r = 0.037	r = 0.243	r = 0.048	r = 0.019	r = 0.220	r = 0.207	r = 0.109
**MEP Amplitude** **115% AMT (µV)**	r = 0.268	r = 0.157	r = 0.080	r = 0.291	r = 0.183	r = 0.294	r = 0.069	r < 0.001	r = 0.197	r = 0.272
**MEP Amplitude** **125% AMT (µV)**	r = 0.352	r = 0.228	r = 0.121	r = 0.243	r = 0.190	r = 0.333	r = 0.012	r = 0.010	r = 0.188	r = 0.252
**MEP Amplitude** **135% AMT (µV)**	r = 0.379	r = 0.332	r = 0.233	r = 0.331	r = 0.342	r = 0.405	r = 0.014	r = 0.109	r = 0.268	r = 0.324
**MEP Amplitude** **145% AMT (µV)**	r = 0.259	r = 0.338	r = 0.214	r = 0.300	r = 0.292	r = 0.402	r = 0.066	r = 0.019	r = 0.287	r = 0.371
**MEP Amplitude** **155% AMT (µV)**	r = 0.264	r = 0.286	r = 0.200	r = 0.390	r = 0.304	r = 0.369	r = 0.006	r = 0.117	r = 0.255	r = 0.388
**eREC AUC**	r = 0.360	r = 0.317	r = 0.221	r = 0.388	r = 0.328	r = 0.403	r = 0.030	r = 0.067	r = 0.300	r = 0.367
**eREC Slope**	r = 0.246	r = 0.246	r = 0.211	r = 0.388	r = 0.274	r = 0.399	r = 0.031	r = 0.082	r = 0.240	r = 0.384
**eREC R^2^**	r = 0.116	r = 0.344	r = 0.049	r = 0.185	r = 0.004	r = 0.242	r = 0.033	r = 0.047	r = 0.148	r = 0.204
**CSP Duration** **105% AMT (ms)**	r = 0.298	r = 0.282	r = 0.120	r = 0.324	r = 0.262	r = 0.396	r = 0.113	r = 0.176	r = 0.166	r = 0.284
**CSP Duration** **115% AMT (ms)**	r = 0.259	r = 0.364	r = 0.136	r = 0.288	r = 0.222	r = 0.287	r = 0.070	r = 0.135	r = 0.111	r = 0.20
**CSP Duration** **125% AMT (ms)**	r = 0.273	r = 0.235	r = 0.116	r = 0.235	r = 0.267	r = 0.295	r = 0.087	r = 0.186	r = 0.098	r = 0.209
**CSP Duration** **135% AMT (ms)**	r = 0.297	r = 0.234	r = 0.015	r = 0.213	r = 0.159	r = 0.272	r = 0.084	r = 0.244	r = 0.008	r = 0.176
**CSP Duration** **145% AMT (ms)**	r = 0.362	r = 0.248	r = 0.060	r = 0.127	r = 0.221	r = 0.215	r = 0.142	r = 0.251	r = 0.039	r = 0.058
**CSP Duration** **155% AMT (ms)**	r = 0.324	r = 0.239	r = 0.051	r = 0.121	r = 0.222	r = 0.244	r = 0.201	r = 0.340	r = 0.001	r = 0.021
**iREC AUC**	r = 0.313	r = 0.279	r = 0.088	r = 0.198	r = 0.228	r = 0.260	r = 0.104	r = 0.229	r = 0.056	r = 0.013
**iSP Onset Latency (ms)**	r = 0.552	r = 0.526	r = 0.237	r = 0.506	r = 0.425	r = 0.471	r = 0.004	r = 0.009	r = 0.194	r = 0.290
**iSP Duration (ms)**	r = 0.330	r = 0.082	r = 0.169	r = 0.065	r = 0.111	r = 0.248	r = 0.077	r = 0.022	r = 0.058	r = 0.062
**iSP AUC**	r = 0.356	r = 0.281	r = 0.289	r = 0.065	r = 0.296	r = 0.173	r = 0.030	r = 0.158	r = 0.034	r = 0.164
**iSP Depth**	r = 0.263	r = 0.270	r = 0.219	r = 0.272	r = 0.270	r = 0.288	r = 0.039	r = 0.215	r = 0.267	r = 0.297
**iSP Depth×Duration**	r = 0.451	r = 0.014	r = 0.248	r = 0.192	r = 0.220	r = 0.368	r = 0.076	r = 0.135	r = 0.187	r = 0.201
Statistically significant findings (sig.) are highlighted as:	Sig. ≤ 0.001	Sig. ≤ 0.010	Sig. ≤ 0.050							

Note: AMT, active motor threshold; AUC, area under the curve; CSP, cortical silent period; eREC, excitatory recruitment curve; iREC, inhibitory recruitment curve; iSP, ipsilateral silent period; MEP, Motor evoked potential; MSO, maximal stimulator output; SDMT, symbol digit modality test; 9HPT, nine-hole peg test. #, absolute r-value reported due to transformation of data.

## Data Availability

All data are available at reasonable request to the corresponding author.

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
