# Peer review of "Probing the Brain–Body Connection Using Transcranial Magnetic Stimulation (TMS): Validating a Promising Tool to Provide Biomarkers of Neuroplasticity and Central Nervous System Function"

_brainsci, 2021, doi:10.3390/brainsci11030384_

Round 1

Reviewer 1 Report

INTRODUCTION

Investigating the functionality of central nervous system circuits with transcranial magnetic stimulation is one of the most advanced frontiers of neurophysiology. Multiple Sclerosis represents only one of the fields of interest of this method since it has been successfully applied also in pathologies classically considered as prevalently involving the peripheral nervous system (see ALS) or even in psychiatric pathologies (see Depression). The knowledge of the protocols achievable through TMS must therefore be deepened because it represents not only a possibility of deepening the preclinical knowledge on the functioning of the CNS but also a field of research for prognostic markers or therapeutic efficacy in neurological diseases.

The study begins with a detailed explanation of the TMS protocols that the investigators have decided to use and also explores their neurophysiological correlations, clarifying which of them investigate excitatory, inhibitory and connective intracortical functions. The precision with which these protocols are illustrated certainly represents a strong point of the article.

The aim of the study is precisely to apply some of the currently best known protocols in a population of patients with multiple sclerosis and investigate the correlation with clinical outcomes of motor and cognitive function. Since the main feature of TMS studies is to assess the functionality of the corticospinal bundle, it is implicit that there are greater correlations with motor outcomes, however in this case, as well as in much of the literature, it is common to find surprising correlations with cognitive-behavioral outcomes. It may be useful to cite some paper that has used TMS to assess the therapeutic effect of several drugs 

Carotenuto A, Iodice R, Petracca M, Inglese M, Cerillo I, Cocozza S, Saiote C, Brunetti A, Tedeschi E, Manganelli F, Orefice G. Upper motor neuron evaluation in multiple sclerosis patients treated with Sativex®. Acta Neurol Scand. 2017 Apr;135(4):442-448. doi: 10.1111/ane.12660. Epub 2016 Aug 8. PMID: 27500463.

RELEVANCE FOR MULTIPLE SCLEROSIS AND PROXIMITY TO CURE

Multiple Sclerosis is the leading cause of disability in the population of young adults in the Western world and has a significant impact on their quality of life. Despite the improvement of disease modifying therapies, it is often impossible to stop its progression and therefore an important part of patient management is neurorehabilitation.
Spasticity, upper limb motor impairment, cognitive slowing and fatigue are daily obstacles for the patient and significantly affect his autonomy although they do not have an adequate relevance in the EDSS scale, the most used in the literature for the assessment of disability in patients with Multiple Sclerosis. The choice to use other clinical outcomes, in addition to the EDSS, represents, therefore, a strength of the study, making it much closer to the daily problems of the patient and therefore to the improvement of quality of life.

The possibility of having useful biomarkers to identify the forms with worse prognosis and to monitor the progress of rehabilitation therapies would involve a radical change in the management of the disease and patients in the long term. If specific neurophysiological alterations and the rehabilitation programs that improve them, could be identified, a personalized rehabilitation program could be created, which is extremely suggestive in such a clinically heterogeneous pathology. It could be useful to cite 

Iodice R, Manganelli F, Dubbioso R. The therapeutic use of non-invasive brain stimulation in multiple sclerosis - a review. Restor Neurol Neurosci. 2017;35(5):497-509. doi: 10.3233/RNN-170735. PMID: 28984619.

MATERIALS AND METHODS

The study aims to identify a protocol of transcranial magnetic stimulation that is both rapid and effective. The TMS protocols, in fact, require numerous measurements of evoked potentials in order to reach significance and it is therefore essential that the patient ensures maximum compliance. For this reason, although the idea of investigating both hemispheres in the same patient is important in a research context, in clinical application it would be more appropriate to focus exclusively on the study of the 'weak' hemisphere since it is the one that has shown the most significant results.

Despite being the gold standard for the assessment of manual dexterity in the literature, the NHPT is a method considered in some studies, too superficial to highlight motor impediments in fine movements. The use of more expensive, but easy-to-use methods such as the hand test system could be used in future studies.

The fatigue described in patients with multiple sclerosis is a condition that significantly limits the patient's autonomy. It may be appropriate, therefore, to combine the walking test with questionnaires on ADL and IADL. These questionnaires indirectly investigate also the motor and cognitive function of the patient, thus allowing to evaluate with a single parameter the general autonomy of the patient.

Finally, the use of recruitment curves is highly encouraged in the literature for its correlation with cortical excitability and manual dexterity. However, it is often necessary to apply stimulations even higher than 150% of the RMT in order to investigate them properly. Stopping at stimuli of lower intensity could significantly alter the results obtained and mask new correlations.

COMMENTARY

The study represents an interesting and stimulating work, especially for the richness of neurophysiological details. As previously stated, the possibility of using TMS to identify prognostic and therapeutic biomarkers could be of great importance for all neurology, as well as for multiple sclerosis.

However, there are methodological flaws that should be corrected.

Multiple Sclerosis is an extremely heterogeneous disease and it is known that about 10% of patients are affected by the primary progressive form. Numerous data in the literature testify that, although having in common the characteristics of demyelinating disease, the primary progressive and relapsing remitting forms differ significantly in genetics, neuropathology, neurophysiology, prognosis and therapy. In the study 5 primary progressive patients are included, which on one hand are few compared to the total to include such a different form of the disease, on the other hand could affect the significance of the data collected for the relapsing remitting population. It would be appropriate, therefore, to exclude these patients from the study and make another in parallel, with comparable sample size, which concerns only the primary progressives.

Moreover, it is impossible to evaluate inhibitory and excitatory neurophysiological parameters of the CNS, without considering the use of drugs that influence them. The effect of neuropsychological drugs, including antidepressants, commonly used in the general population and extremely frequent in patients with MS, is able to alter the electrophysiological patterns of the CNS (in the literature is increasingly advancing the concept of drug-TMS). Including in the same sample patients who use these drugs, irreparably alters the reliability of the study.

The paired-pulse parameters of TMS represent the main future perspective of deepening the study. They are parameters that investigate intracortical excitatory and inhibitory circuits and it is possible that some of them have even more  correlations with motor or cognitive outcomes. The strategy to start from single-pulse protocols is anyway correct, since they are of shorter duration and therefore require less compliance from the patient.

Author Response

All changes in the manuscript are highlighted in red.

-------------------------------------------------------------------------------

It may be useful to cite some paper that has used TMS to assess the therapeutic effect of several drugs. Carotenuto A, Iodice R, Petracca M, Inglese M, Cerillo I, Cocozza S, Saiote C, Brunetti A, Tedeschi E, Manganelli F, Orefice G. Upper motor neuron evaluation in multiple sclerosis patients treated with Sativex®. Acta Neurol Scand. 2017 Apr;135(4):442-448. doi: 10.1111/ane.12660. Epub 2016 Aug 8. PMID: 27500463.

We have included the suggested reference in the manuscript (introduction, line 39, and 177).

The possibility of having useful biomarkers to identify the forms with worse prognosis and to monitor the progress of rehabilitation therapies would involve a radical change in the management of the disease and patients in the long term. If specific neurophysiological alterations and the rehabilitation programs that improve them, could be identified, a personalized rehabilitation program could be created, which is extremely suggestive in such a clinically heterogeneous pathology. It could be useful to cite

Iodice R, Manganelli F, Dubbioso R. The therapeutic use of non-invasive brain stimulation in multiple sclerosis - a review. Restor Neurol Neurosci. 2017;35(5):497-509. doi: 10.3233/RNN-170735. PMID: 28984619.

We have added the suggested reference to the manuscript (introduction, lines 151-152)

MATERIALS AND METHODS

Despite being the gold standard for the assessment of manual dexterity in the literature, the NHPT is a method considered in some studies, too superficial to highlight motor impediments in fine movements. The use of more expensive, but easy-to-use methods such as the hand test system could be used in future studies.

              We agree with the reviewer that the 9HPT is the gold standard to assess manual dexterity impairments. At the moment there is no better validated tool.

Finally, the use of recruitment curves is highly encouraged in the literature for its correlation with cortical excitability and manual dexterity. However, it is often necessary to apply stimulations even higher than 150% of the RMT in order to investigate them properly. Stopping at stimuli of lower intensity could significantly alter the results obtained and mask new correlations.

               We agree with the reviewer, and we have also encouraged such methodology in the manuscript (e.g. see figure 2). Our experience with higher stimulations intensities in this population though is that, especially for more disabled participants, going over 155% of AMT can often require MSO% that are higher than 100%. Also, our aim was to keep the number of stimulations reduced to avoid participant burden during TMS assessment.

COMMENTARY

In the study 5 primary progressive patients are included, which on one hand are few compared to the total to include such a different form of the disease, on the other hand could affect the significance of the data collected for the relapsing remitting population. It would be appropriate, therefore, to exclude these patients from the study and make another in parallel, with comparable sample size, which concerns only the primary progressives.

              The reviewer makes a great point. In fact, we have taken this approach in previous TMS work from our lab (see Chaves, et al (2019), Clinical Neurophysiology). We plan in the future, once we have enough participants to allow us for comparison, to investigate CSE differences across MS types. The aim of this present research, however, was firstly, to describe the TMS method, and secondly, determine whether this method predicts symptom severity. Having a wider range of impairment and disability scores, and therefore, having these PPMS patients included was considered important.

Moreover, it is impossible to evaluate inhibitory and excitatory neurophysiological parameters of the CNS, without considering the use of drugs that influence them. The effect of neuropsychological drugs, including antidepressants, commonly used in the general population and extremely frequent in patients with MS, is able to alter the electrophysiological patterns of the CNS (in the literature is increasingly advancing the concept of drug-TMS). Including in the same sample patients who use these drugs, irreparably alters the reliability of the study.

              We agree with the reviewer. Many of our participants, on top of their prescribed disease-modifying therapies, were prescribed other medications (often more than one). Therefore, our future research will examine in more detail the use of CNS-modulating and disease-modifying drugs and their impact on CSE in MS. We believe that this important topic deserves a more in-depth attention and its separate research design and analysis. However, to consider the reviewer’s point and provide some insight for future research investigating drugs and CSE in MS, an approach we have taken now was to collect participants’ medication and separate them into two major groups: 1) excitatory and 2) Inhibitory modulating CNS drugs. We have then performed a Chi-square / Fisher Exact analysis to compare whether both groups would have a different proportion of more or less disabled participants using CNS excitatory and inhibitory modulating-drugs. Whereas for excitatory drugs, the proportion did not differ between more and less disabled participants, there was a higher proportion of more disabled participants undergoing prescription of inhibitory-modulating drugs. We have reported this new analysis and its results in the new manuscript, and based on these results we have argued that (discussion, limitation section, lines 965-972) ‘It is important to appreciate that drugs that modulate the CNS likely influence CSE. We report that in this cohort, inhibitory (but not excitatory) modulating CNS drugs were more frequently prescribed among participants with higher levels of disability. Changes in TMS biomarkers of cortical inhibition (e.g. CSP) in MS could be related, in part, to medications (e.g. for management of chronic pain, spasticity, sleep issues, anxiety, bowel and bladder issues). Research that focus primarily on elucidating the effects of drugs on the CSE of people with MS is needed.’

Reviewer 2 Report

Summary

The authors present a highly detailed and explanative description of various TMS variable findings in PwMS. Despite the length and detail of the manuscript it reads well and is easy to follow. Thus, I only have a few relatively minor suggestions for strengthening the manuscript.

Introduction

Major comments

  • Lines 60 – 62: Please provide a citation for excitation being mediated by glutamate and inhibition being mediated by GABA.
  • Line 89: What does “accuracy of corticospinal neurons… (R2)” mean? A clear description will bring this variable on par with the rest of the Introduction.

Materials and Methods

Minor comments

  • Lines 472 – 473: “…Wilcoxon Signed-Ranks test (for data non-normally distributed data,…) has some redundancy. Please revise for clarity.

Results

Major comments

  • Table 1: Because many of your results are driven by the subjects with more disability, you should report the subgroup characteristics (more disability and less disability). I see that you do this already in Tables 2 and 3, to some degree, but it would be more informative to present the complete characteristics here.
  • Table 1: To be complete, you should also describe the acronym “9HPT” in the table note.
  • Lines 541 – 544: I think these are descriptions for data in Figure 8, not Figure 7.
  • Figures 7 – 9: What do the error bars in the graphs represent? SD or SEM?

Discussion

Major comments

  • Line 888: Is MEP latency influenced by muscle activity status (RMT vs. AMT) during MEP determination? Could this partially explain the differences between your results and those others you present?

Conclusion

Major comment: I think you need to say something about differences varying by disease severity. It is a prominent aspect of your Results and bears mention in the Conclusion. Also, I think it would be helpful to explicitly readdress your purposes/research questions found at the end of the Introduction and structure your Conclusion accordingly. E.g., did you meet your goals? Specifically, based on the results of your study, can you list the “core set” of variables you might recommend? I appreciate that you do this already to some extent, but a summary statement, (e.g., “Thus, we recommend that 1) iSP, 2)…, and X) XXX be included as a minimum core set for future TMS investigations.”) would clearly summarize your results in the context of your stated intentions.

Author Response

Changes in the new manuscript are highlighted in red

--------------------------------------------------------------

Lines 60 – 62: Please provide a citation for excitation being mediated by glutamate and inhibition being mediated by GABA.

              Two citations were added to the manuscript (provided below).

Ziemann, U.; Reis, J.; Schwenkreis, P.; Rosanova, M.; Strafella, A.; Badawy, R.; Muller-Dahlhaus, F. TMS and drugs revisited 2014. Clinical neurophysiology : official journal of the International Federation of Clinical Neurophysiology 2015, 126, 1847-1868, doi:10.1016/j.clinph.2014.08.028.

Rossini, P.M.; Burke, D.; Chen, R.; Cohen, L.G.; Daskalakis, Z.; Di Iorio, R.; Di Lazzaro, V.; Ferreri, F.; Fitzgerald, P.B.; George, M.S., et al. Non-invasive electrical and magnetic stimulation of the brain, spinal cord, roots and peripheral nerves: Basic principles and procedures for routine clinical and research application. An updated report from an I.F.C.N. Committee. Clinical neurophysiology : official journal of the International Federation of Clinical Neurophysiology 2015, 126, 1071-1107, doi:10.1016/j.clinph.2015.02.001.

Line 89: What does “accuracy of corticospinal neurons… (R2)” mean? A clear description will bring this variable on par with the rest of the Introduction.

              We added new text and provided references to support this sentence. It reads (lines 92-96): ‘Previous work has demonstrated reduced gain (eREC slope) and overall excitability (eREC AUC) in stroke survivors, as well as further associations between these eREC parameters with central nervous system damage beyond the primary motor area [19]. As for accuracy (eREC R2), previous authors have proposed that values below 0.7 may reflect poor ability of the brain in recruiting neurons accurately [20,21].’

[20] Potter-Baker, K.A.; Varnerin, N.M.; Cunningham, D.A.; Roelle, S.M.; Sankarasubramanian, V.; Bonnett, C.E.; Machado, A.G.; Conforto, A.B.; Sakaie, K.; Plow, E.B. Influence of Corticospinal Tracts from Higher Order Motor Cortices on Recruitment Curve Properties in Stroke. Frontiers in neuroscience 2016, 10, 79-79, doi:10.3389/fnins.2016.00079.

[21] Carson, R.G.; Nelson, B.D.; Buick, A.R.; Carroll, T.J.; Kennedy, N.C.; Cann, R.M. Characterizing Changes in the Excitability of Corticospinal Projections to Proximal Muscles of the Upper Limb. Brain stimulation 2013, 6, 760-768, doi:https://doi.org/10.1016/j.brs.2013.01.016.

Lines 472 – 473: “…Wilcoxon Signed-Ranks test (for data non-normally distributed data,…) has some redundancy. Please revise for clarity.

              We have corrected this sentence and removed this duplicate (‘data’).

Table 1: Because many of your results are driven by the subjects with more disability, you should report the subgroup characteristics (more disability and less disability). I see that you do this already in Tables 2 and 3, to some degree, but it would be more informative to present the complete characteristics here.

              Demographic data from these two groups are now reported in table 1.

Table 1: To be complete, you should also describe the acronym “9HPT” in the table note.

              We have added this description to the table note.

Lines 541 – 544: I think these are descriptions for data in Figure 8, not Figure 7.

              The reviewer is correct. This has now been fixed.

Figures 7 – 9: What do the error bars in the graphs represent? SD or SEM?

              The bars represent one SD. We have now reported this in all figures’ legends.

Line 888: Is MEP latency influenced by muscle activity status (RMT vs. AMT) during MEP determination? Could this partially explain the differences between your results and those others you present?

              Yes. Previous research has demonstrated that contracting MEP latencies are shorter. However, the reason why this phenomenon occurs is still unknown. It was not the aim of our research to investigate whether latencies would differ between the two conditions (relaxing vs active). However, we agree with the reviewer that this might be an interesting discussion that might be worth of future investigations. We have added new text to the discussion. It reads (lines 953-958): “Furthermore. MEP latency is influenced by muscle activity (resting vs active). MEPs derived from contracting muscle have shorter MEP latencies (~2 ms faster) for reasons that are yet to be determined [12]. Since MEPs derived from contracted and resting muscles are likely governed by different brain structures and processes [152,153], future research should attempt to understand MEP latency differences in clinical populations].”

I think you need to say something about differences varying by disease severity. It is a prominent aspect of your Results and bears mention in the Conclusion.

              The following has been added to the Conclusions section “Indeed, although this asymmetry was not detectable in people with milder MS (EDSS<2.5), the differences in excitability between hemispheres became more apparent once EDSS reached 3.0. “. Furthermore, based on feedback from Reviewer 3, further analysis of low and high disability groups in terms of prescription and recreational drugs was undertake.

Also, I think it would be helpful to explicitly readdress your purposes/research questions found at the end of the Introduction and structure your Conclusion accordingly. E.g., did you meet your goals? Specifically, based on the results of your study, can you list the “core set” of variables you might recommend? I appreciate that you do this already to some extent, but a summary statement, (e.g., “Thus, we recommend that 1) iSP, 2)…, and X) XXX be included as a minimum core set for future TMS investigations.”) would clearly summarize your results in the context of your stated intentions.

              As per reviewer’s suggestion, we have included this summary statement and presented it as a very last result’s section (3.12) on the manuscript (lines 713-732).

Reviewer 3 Report

This manuscript presents the largest cohort of people with Multiple Sclerosis (MS) studied with TMS to date. The authors used TMS-EMG measures, both related to cortical excitability and inhibition. The manuscript has several important and novel findings.

  1. Delayed and longer iSP consistently predicted slower walking speed and poorer hand function.
  2. Longer cortical silent period was the most robust predictor of fatigue, while higher RMT was the best predictor of slower cognitive processing speed.
  3. Greater interhemispheric asymmetry of participants’ corticospinal excitability (measured using AMT) was significantly correlated with overall poorer performance in the greatest number of clinical outcomes.
  4. Values derived from the hemisphere corresponding to the weaker hand resulted in the strongest relationships to clinical outcomes.

Overall, the manuscript reviews the essential previous literature, presents a rigorous study design, and the results support the conclusions. I only have a minor suggestion. For clarity, the authors should define all the abbreviations in their figures. Currently, many of the figures have abbreviations to which the reader needs to find the correct definition from the main text.

Author Response

Changes in the new manuscript are highlighted in red

---------------------------------------------------------------

For clarity, the authors should define all the abbreviations in their figures. Currently, many of the figures have abbreviations to which the reader needs to find the correct definition from the main text.

              Thank you. We have now defined all abbreviations in the figures’ legends.